# On the role of nucleotides and lipids in the polymerization of the actin homolog MreB from a Gram-positive bacterium

Wei Mao[1], Lars D Renner[2]*, Charlène Cornilleau[1], Ines Li de la Sierra-Gallay[3], Sana Afensiss[1], Sarah Benlamara[1], Yoan Ah-Seng[1], Herman Van Tilbeurgh[3], Sylvie Nessler[3]*, Aurélie Bertin[4]*, Arnaud Chastanet[1]*, Rut Carballido-Lopez[1]*

[1]Université Paris-Saclay, INRAE, AgroParisTech, Micalis Institute, Jouy-en-Josas, France; [2]Leibniz Institute of Polymer Research, and the Max-Bergmann-Center of Biomaterials, Dresden, Germany; [3]Institute for Integrative Biology of the Cell (I2BC), Université Paris-Saclay, CEA, CNRS, Gif-sur-Yvette, France; [4]Laboratoire Physico Chimie Curie, Institut Curie, PSL Research University, CNRS UMR168, Sorbonne Université, 75005, Paris, France

*For correspondence: renner@ipfdd.de (LDR); Sylvie.NESSLER@i2bc.paris-saclay.fr (SN); aurelie.bertin@curie.fr (AB); arnaud.chastanet@inrae.fr (AC); rut.carballido-lopez@inrae.fr (RC-L)

Competing interest: The authors declare that no competing interests exist.

**Abstract** *In vivo,* bacterial actin MreB assembles into dynamic membrane-associated filamentous structures that exhibit circumferential motion around the cell. Current knowledge of MreB biochemical and polymerization properties *in vitro* remains limited and is mostly based on MreB proteins from Gram-negative species. In this study, we report the first observation of organized protofilaments by electron microscopy and the first 3D-structure of MreB from a Gram-positive bacterium. We show that *Geobacillus stearothermophilus* MreB forms straight pairs of protofilaments on lipid surfaces in the presence of ATP or GTP, but not in the presence of ADP, GDP or non-hydrolysable ATP analogs. We demonstrate that membrane anchoring is mediated by two spatially close short hydrophobic sequences while electrostatic interactions also contribute to lipid binding, and show that the population of membrane-bound protofilament doublets is in steady-state. In solution, protofilament doublets were not detected in any condition tested. Instead, MreB formed large sheets regardless of the bound nucleotide, albeit at a higher critical concentration. Altogether, our results indicate that both lipids and ATP are facilitators of MreB polymerization, and are consistent with a dual effect of ATP hydrolysis, in promoting both membrane binding and filaments assembly/disassembly.

## Editor's evaluation

This important study makes the case that the assembly of MreB from Geobacillus, a Gram-positive organism differs substantially from MreB from the Gram-negative model organism, *Escherichia coli*. They make the compelling case that Geobacillus MreB assembly requires both interactions with membrane lipids and nucleotide binding: nucleotide hydrolysis is required for interaction with the membrane and interaction with lipids triggers polymerization. Altogether, these data make the strong case that MreB assembly dynamics can vary in significant, and organism-specific ways.

## Introduction

Cytoskeletal proteins are known to polymerize into filaments that play critical roles in various aspects of cell physiology, including cell shape, mechanical strength and motion, cytokinesis, chromosome partitioning and intracellular transport. Prokaryotic cells contain homologs of the main eukaryotic

cytoskeletal proteins, namely actin, tubulin and intermediate filaments (*Cabeen and Jacobs-Wagner, 2010*; *Lin and Thanbichler, 2013*; *Shaevitz and Gitai, 2010*), which were identified decades after their eukaryotic counterparts. In 2001, MreB proteins of the Gram-positive model bacterium *Bacillus subtilis* were found to form filamentous structures underneath the cytoplasmic membrane and to play a key role in the determination and maintenance of rod-shape (*Carballido-Lopez, 2017*; *Jones et al., 2001*). Soon after, the three-dimensional structure of one of the two MreB isoforms from the Gram-negative thermophilic bacterium *Thermotoga maritima* (MreB$^{Tm}$) was solved (*van den Ent et al., 2001*), confirming its structural homology with actin (*Bork et al., 1992*). Besides, MreB$^{Tm}$ in solution was shown to assemble into filaments similar to filamentous actin (F-actin; *van den Ent et al., 2001*).

Research in the field of eukaryotic actin historically focused on elucidating structure-function relationships from *in vitro* studies. The availability of large amounts of soluble actin purified from several cell types since the 1940s enabled decades of mechanistic studies on actin polymerization (*Pollard, 2016*). In contrast, MreB from mesophilic bacteria proved particularly difficult to purify in a soluble form, thwarting efforts to perform *in vitro* assays. Instead, research on MreB primarily focused on cellular studies, driven by the advent of fluorescent microscopy in bacterial cell biology. Over the past two decades, the subcellular localization and dynamics of MreB have been described in several Gram-negative and Gram-positive species (*Billaudeau et al., 2017*; *Billaudeau et al., 2019*; *Dion et al., 2019*; *Harris et al., 2014*; *Hussain et al., 2018*; *Olshausen et al., 2013*; *Oswald et al., 2016*; *Ouzounov et al., 2016*; *Renner et al., 2013*; *Schirner et al., 2015*). *In vivo*, MreB proteins form discrete, nanometer-long, membrane-associated polymeric assemblies along the cell cylinder that move processively around the rod circumference together with proteins of the cell wall (CW) elongation machinery (*Domínguez-Escobar et al., 2011*; *Garner et al., 2011*; *van Teeffelen et al., 2011*), forming the so-called Rod complex. The motility of the Rod complex is driven by CW synthesis (*Domínguez-Escobar et al., 2011*; *Garner et al., 2011*; *van Teeffelen et al., 2011*) and MreB assemblies self-align circumferentially, along their direction of motion (*Billaudeau et al., 2019*; *Hussain et al., 2018*). Recently, it was proposed that the specific intrinsic curvature of MreB polymers increases their affinity for the greatest concave (negative) membrane curvature within the cell, that is the inner surface of the rod circumference, accounting for their orientation (*Hussain et al., 2018*). The current model is that self-aligned MreB filaments restrict the diffusion of CW biosynthetic proteins in the membrane and orient their motion to insert new peptidoglycan strands in radial hoops perpendicular to the long axis of the cell, promoting the cylindrical expansion of rod-shaped cells (*Domínguez-Escobar et al., 2011*; *Garner et al., 2011*; *Hussain et al., 2018*). However, many questions remain to be answered. What prompts the assembly of MreB on the inner leaflet of the cytoplasmic membrane? What is the architecture of the membrane-associated MreB polymeric assemblies and how is it controlled? How is their distribution along the cell cylinder regulated? What is the length of individual MreB filaments within these assemblies and how is it controlled? Are the filaments stable? Do they exhibit turnover like actin filaments? *In vivo*, the length of MreB filamentous assemblies can be affected by the intracellular concentration of the protein (*Billaudeau et al., 2019*; *Salje et al., 2011*), but seems to have little impact on MreB function (*Billaudeau et al., 2019*). No turnover of MreB assemblies was detected *in vivo*, at least relative to their motion around the cell circumference (*Domínguez-Escobar et al., 2011*; *van Teeffelen et al., 2011*). Therefore, MreB polymers are believed to be quite stable despite their motion in the cell. To elucidate in detail the molecular mechanisms underlying the functions of MreB, it remains necessary to understand their biochemical and polymerization properties. The majority of biochemical and structural studies on MreB proteins originally focused on the highly soluble Gram-negative MreB$^{Tm}$ (*Bean and Amann, 2008*; *Esue et al., 2005*; *Esue et al., 2006*; *Popp et al., 2010b*; *van den Ent et al., 2001*; *van den Ent et al., 2010*). The tendency to aggregation upon purification hampered most *in vitro* studies of MreBs from other species. Over the last decade, purification and polymerization assays were nevertheless reported for MreBs from several Gram-negative bacteria, from the Gram-positive *B. subtilis* (MreB$^{Bs}$) and from wall-less *Chlamydophila* and *Spiroplasma* species (*Dersch et al., 2020*; *Gaballah et al., 2011*; *Harne et al., 2020*; *Maeda et al., 2012*; *Mayer and Amann, 2009*; *Nurse and Marians, 2013*; *Pande et al., 2022*; *Salje et al., 2011*; *Takahashi et al., 2022*; *van den Ent et al., 2014*).

Direct binding to the cell membrane was shown for MreB from the Gram-negative *Escherichia coli* (MreB$^{Ec}$) and *T. maritima* (*Salje et al., 2011*) and, more recently, for MreB from *Spiroplasma citri* (MreB5$^{Sc}$; *Harne et al., 2020*). The N-terminal amphipathic helix of MreB$^{Ec}$ was found to be necessary

for membrane binding and also to cause the full-length purified protein to aggregate (*Salje et al., 2011*). Although this N-terminal amphipathic helix is dispensable for polymerization, it is required for proper function of MreB[Ec] *in vivo* (*Salje et al., 2011*). MreB[Tm] and MreB5[Sc] are devoid of such an N-terminal amphipathic helix, but instead possess a small hydrophobic loop that protrudes from the monomeric globular structure and mediates membrane binding (*Pande et al., 2022*; *Salje et al., 2011*). Additionally, an acidic C-terminal tail was shown to mediate a charge-based interaction of MreB5[Sc] with the membrane (*Pande et al., 2022*).

Altogether, *in vitro* work on MreBs from Gram-negative bacteria has shown that MreB polymerizes into straight double filaments in the presence of nucleotides, both in solution and on lipid membrane surfaces (*Harne et al., 2020*; *Salje et al., 2011*; *Takahashi et al., 2022*; *van den Ent et al., 2014*; *van den Ent et al., 2010*), and that filaments can assemble into larger sheets by lateral interactions (*Esue et al., 2005*; *Esue et al., 2006*; *Harne et al., 2020*; *Nurse and Marians, 2013*; *Popp et al., 2010b*; *van den Ent et al., 2001*; *van den Ent et al., 2014*). Furthermore, work on *Caulobacter crescentus* MreB (MreB[Cc]) and MreB[Ec] indicated an antiparallel arrangement of the straight pairs of protofilaments (*van den Ent et al., 2014*), in sharp contrast to the helical parallel pairs of protofilaments (double helix) characteristic of F-actin (*Pollard, 1990*). While the parallel arrangement of a protofilament doublet generates polarity and allows for the characteristic treadmilling of F-actin (*Stoddard et al., 2017*), the antiparallel arrangement in MreB protofilaments suggests a bidirectional polymerization/depolymerization mechanism (*van den Ent et al., 2014*). The directionality and the kinetics of MreB polymerization, as well as the role of nucleotides in this process remain to be shown. ATPase activity has been reported in solution for MreB[Tm], MreB[Ec], MreB[Bs] and *Spiroplasma* MreBs (*Esue et al., 2005*; *Esue et al., 2006*; *Mayer and Amann, 2009*; *Nurse and Marians, 2013*; *Pande et al., 2022*; *Popp et al., 2010b*; *Takahashi et al., 2022*). However, the need for nucleotide binding and hydrolysis in polymerization remains unclear. Early reports suggested a strict dependency on hydrolysable nucleotides (ATP or GTP) for polymerization of MreB[Tm] (*Esue et al., 2006*; *van den Ent et al., 2001*), and later for MreB[Ec] (*Nurse and Marians, 2013*), while others claimed that polymerization occurred similarly in the presence of ADP (*Bean and Amann, 2008*; *Gaballah et al., 2011*; *Mayer and Amann, 2009*; *Pande et al., 2022*; *Popp et al., 2010b*; *Takahashi et al., 2022*), of the non-hydrolysable ATP analogue AMP-PNP (adenylyl-imidodiphosphate) (*Bean and Amann, 2008*; *Mayer and Amann, 2009*; *Pande et al., 2022*; *Salje et al., 2011*; *Takahashi et al., 2022*), or even in the absence of nucleotide (*Mayer and Amann, 2009*).

No electron microscopy (EM) images of protofilaments or atomic views of MreB from a Gram-positive bacterium have been reported to date. The two *in vitro* studies so far reported on *B. subtilis* MreB investigated polymerization by light scattering and sedimentation assays (*Mayer and Amann, 2009*) and by fluorescence microscopy (*Dersch et al., 2020*), respectively, but provided no evidence that MreB[Bs] polymerized into protofilaments in the conditions tested. In Gram-positive bacteria, MreB proteins presumably have no N-terminal amphipathic helix (*Salje et al., 2011*), and the genome usually encodes several MreB isoforms (in contrast to Gram-negative that usually get by with a single *mreB* paralog), that may be related to their thicker and more complex CW structure (*Chastanet and Carballido-Lopez, 2012*). Inter- and intra-species differences in MreBs may exist at the structural or biochemical level, leading to differences in molecular interactions or biological functions.

In this study, we aimed to decipher fundamental structural and biochemical properties of MreB from a Gram-positive bacterium. We successfully purified a soluble form of MreB from the Gram-positive thermophilic *Geobacillus stearothermophilus* (MreB[Gs]) and elucidated its crystal structure, confirming the classical actin/MreB fold and the presence of the small hydrophobic loop shown to mediate membrane binding in MreB[Tm] and MreB5[Sc] (*Pande et al., 2022*; *Salje et al., 2011*). Polymerization assays showed that MreB[Gs] forms straight pairs of protofilaments in the presence of lipids and nucleotide triphosphate (either ATP or GTP), and that these are dynamic. We also show that the interaction with lipids is mediated by electrostatic interactions (*Pande et al., 2022*) and by two spatially close hydrophobic motifs in the MreB[Gs] monomers that comprise the small hydrophobic loop and the N-terminal end. Free in solution, MreB[Gs] assembled into large sheets regardless of the bound nucleotide, albeit at a higher MreB[Gs] concentration than the one required for polymerization into pairs of protofilaments on a lipid surface. Taken together, our results show a key role for ATP as facilitator of MreB polymerization on the membrane, and suggest that ATP hydrolysis promotes both MreB membrane binding and filament assembly/disassembly.

## Results

### Crystal structure of *G. stearothermophilus* MreB

To solve the structure of a MreB protein from a Gram-positive bacterium but overcome the notorious aggregation issues of MreB from mesophilic bacteria, we cloned and purified MreB from the thermophilic *G. stearothermophilus* (MreB[Gs]). We chose *G. stearothermophilus* because of its proximity to the *Bacillus* genus and because of the highly conserved sequence of MreB[Gs] compared to MreB from the model Gram-positive bacterium *B. subtilis*. MreB[Bs] is more closely related to MreB[Gs] (85.6% identity and 92.6% similarity) than to MreB of Gram-negative for which biochemical or structural data are available (either the thermophilic *T. maritima* with 55.8% identity, or the mesophilic *C. crescentus*, 56.9% identity and *E. coli*, 55.2% identity) (*Figure 1—figure supplement 1*).

MreB[Gs] was purified to homogeneity following a two-step procedure (see Materials and methods). The protein could be purified in a soluble form (*Figure 1—figure supplement 2*) that remained functional for polymerization at concentrations below 13.4 µM (0.5 mg/mL). When stored frozen at higher concentrations or when conserved overnight at 4 °C, MreB[Gs] rapidly aggregated (*Figure 1—figure supplement 2A*) and could not be recovered in a monomeric state, consistent with the known tendency of MreB proteins to aggregate.

The purified MreB[Gs] protein was crystallized and the structure of its apo form was solved at 1.8 Å resolution (Protein Data Bank Identifier (PDB ID) 7ZPT). The crystals belong to the monoclinic $P2_1$ space group and contain one molecule of MreB[Gs] per asymmetric unit (*Supplementary file 1*). Monomers of apo MreB[Gs] display the canonical fold of actin-like proteins, characterized by four subdomains IA, IIA, IB and IIB (*Figure 1A*). One of the most similar structures to apo MreB[Gs] is the apo form of MreB[Tm] (PDB ID 1JCF; *van den Ent et al., 2001*), with a root mean square deviation (rmsd) of 1.92 Å over 305 superimposed Cα atoms and a Z-score of 16.0. Superimposition of the two proteins (*Figure 1A*) revealed that MreB[Gs] is in a slightly more open conformation than MreB[Tm], mainly due to a movement of domain IB, which is the less conserved within the actin superfamily of proteins. Loop β6-α2, which connects subdomains IA and IB and closes the nucleotide-binding pocket, is partially disordered in apo MreB[Gs]. In domain IA, the hydrophobic loop α2-β7, which has been shown to be involved in MreB[Tm] membrane binding (*Salje et al., 2011*) and is 2 residues longer in MreB[Gs] (*Figure 1—figure supplement 1*), displays a distinct conformation, packed on the N-terminal extremity of the polypeptide chain.

Crystal packing analysis revealed straight protofilaments characterized by a subunit repeat distance of 51 Å (*Figure 1B*), similar to that observed in crystal structures of other actin homologs (*Harne et al., 2020*; *Pande et al., 2022*; *Roeben et al., 2006*; *van den Ent et al., 2014*). However, because of the open conformation of MreB[Gs] (*Figure 1A*), domain IB interacts with domain IA (*Figure 1C*) and not with domain IIA as observed for example for MreB[Tm] (*Figure 1D*; *van den Ent et al., 2001*).

### MreB[Gs] polymerizes into straight pairs of protofilaments in the presence of lipids

We next investigated the polymerization of MreB[Gs] by EM of negatively stained samples. *In vivo*, MreB[Bs] forms membrane-associated nanofilaments (*Billaudeau et al., 2019*; *Hussain et al., 2018*; *Jones et al., 2001*), and MreB filaments from Gram-negative bacteria have been shown to have an intrinsic affinity for membranes (*Garenne et al., 2020*; *Maeda et al., 2012*; *Salje et al., 2011*; *van den Ent et al., 2014*). We hypothesized that the presence of lipids might be important for the assembly of MreB[Gs] polymers, and thus performed polymerization reactions in the presence and in the absence of lipids. In the presence of ATP, MreB[Gs] readily formed pairs of protofilaments on a monolayer of total *E. coli* lipid extract, while these were virtually not observed in the absence of lipids (*Figure 2A*). Using a semi-quantitative workflow analysis of TEM grids (*Figure 2—figure supplement 1*; Materials and methods), we found that in the presence of both ATP and lipids, MreB[Gs] formed a lawn of double protofilaments in 100% of fields, while only 4% and 8% of the fields contained polymers (often at very low density) in the absence of either ATP or lipids, respectively (*Figure 2A*). To test if polymers had formed in solution but failed to bind to the hydrophobic EM grids, we instead used glow-discharged hydrophilic grids, which are commonly used to adsorb soluble proteins. Again, MreB[Gs] filaments were not significantly detected in solution (*Figure 2—figure supplement 2A*). In the presence of lipids, the frequency of polymers was drastically reduced on the glow-discharged grids, consistent with impaired adhesion of a lipid monolayer to a hydrophilic surface (*Figure 2—figure supplement 2A*). We next

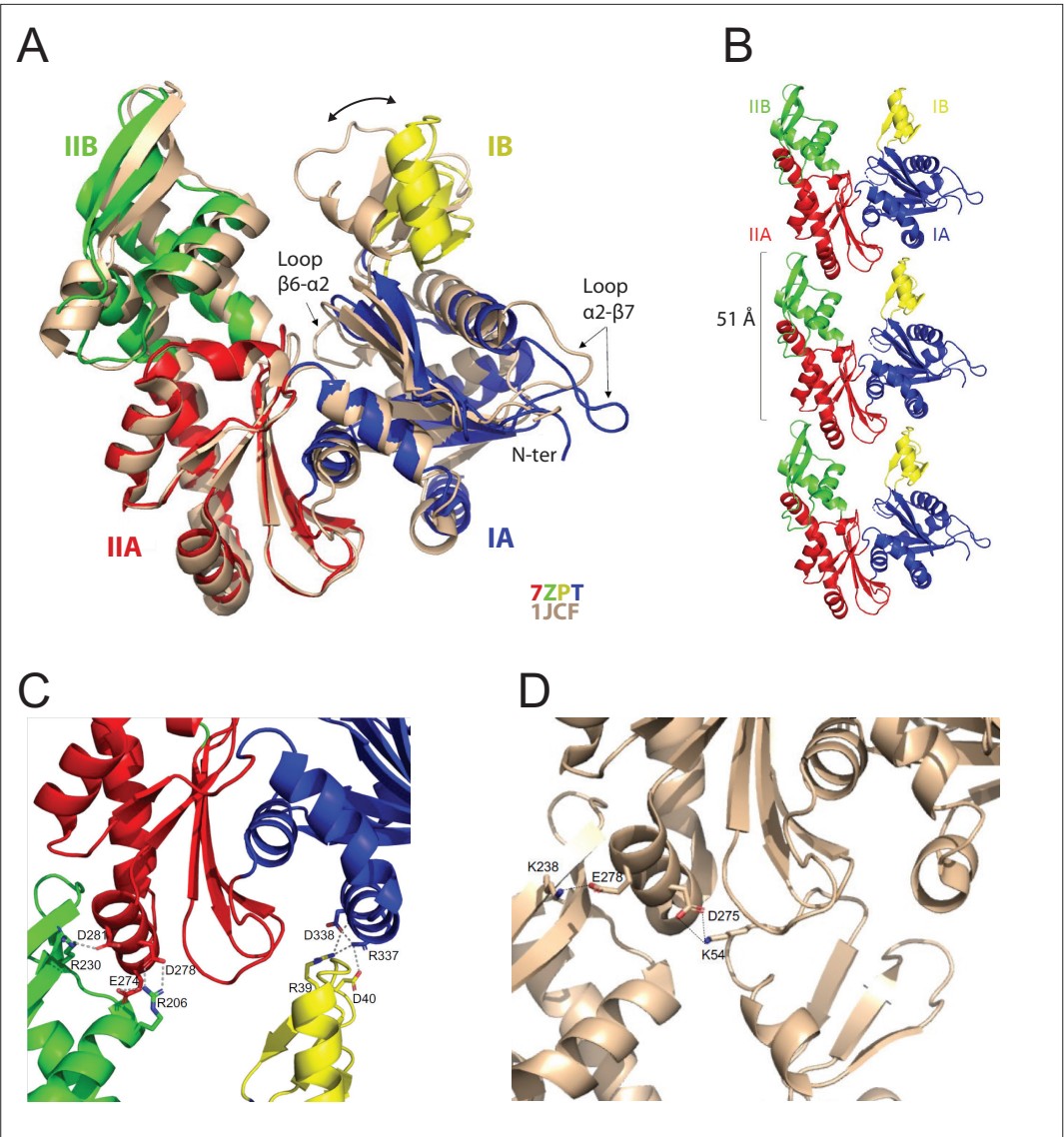

**Figure 1.** Crystal structure of the apo protofilament of MreB from *G. stearothermophilus*. (**A**) Crystal structure of apo MreB^Gs (PDB ID 7ZPT), colored by subdomains, superimposed on the crystal structure of apo MreB^Tm (PDB ID 1JCF), in beige. The sequence similarity between the two proteins is 55.8%. Subdomain IA (blue) of MreB^Gs is formed by residues 1–32, 66–145 and 315–347; subdomain IB (yellow) by residues 33–65; IIA (red) by residues 146–181 and 246–314 and IIB (green) by residues 182–245. Superimposition of the two forms highlights the distinct positions of loops β6-α2 and α2-β7 as well as the movement of domain IB (two-headed arrow) resulting in slightly distinct subunit interaction modes as shown in panel C. (**B**) Protofilament structure of apo MreB^Gs. Three subunits of the protofilament formed upon crystal packing are displayed as cartoon and colored by subdomains. The subunit repeat distance is indicated. (**C**) Close view of the MreB^Gs intra-protofilament interface. The two subunits are colored by subdomains as in panel A, and shown as cartoons. Residues involved in putative salt bridges (gray dashed lines) are displayed as sticks colored by atom type (N in blue and O in red) and labeled. (**D**) Close view of the MreB^Tm intra-protofilament interface (PDB ID 1JCF). The two subunits are colored in beige as in panel A, and shown as cartoons. Residues involved in putative salt bridges (gray dashed lines) are displayed as sticks colored by atom type (N in blue and O in red) and labeled.

The online version of this article includes the following figure supplement(s) for figure 1:

**Figure supplement 1.** Multiple sequence alignment of MreB proteins from several bacteria.

**Figure supplement 2.** Calibration curve and typical size exclusion chromatography profiles of MreB.

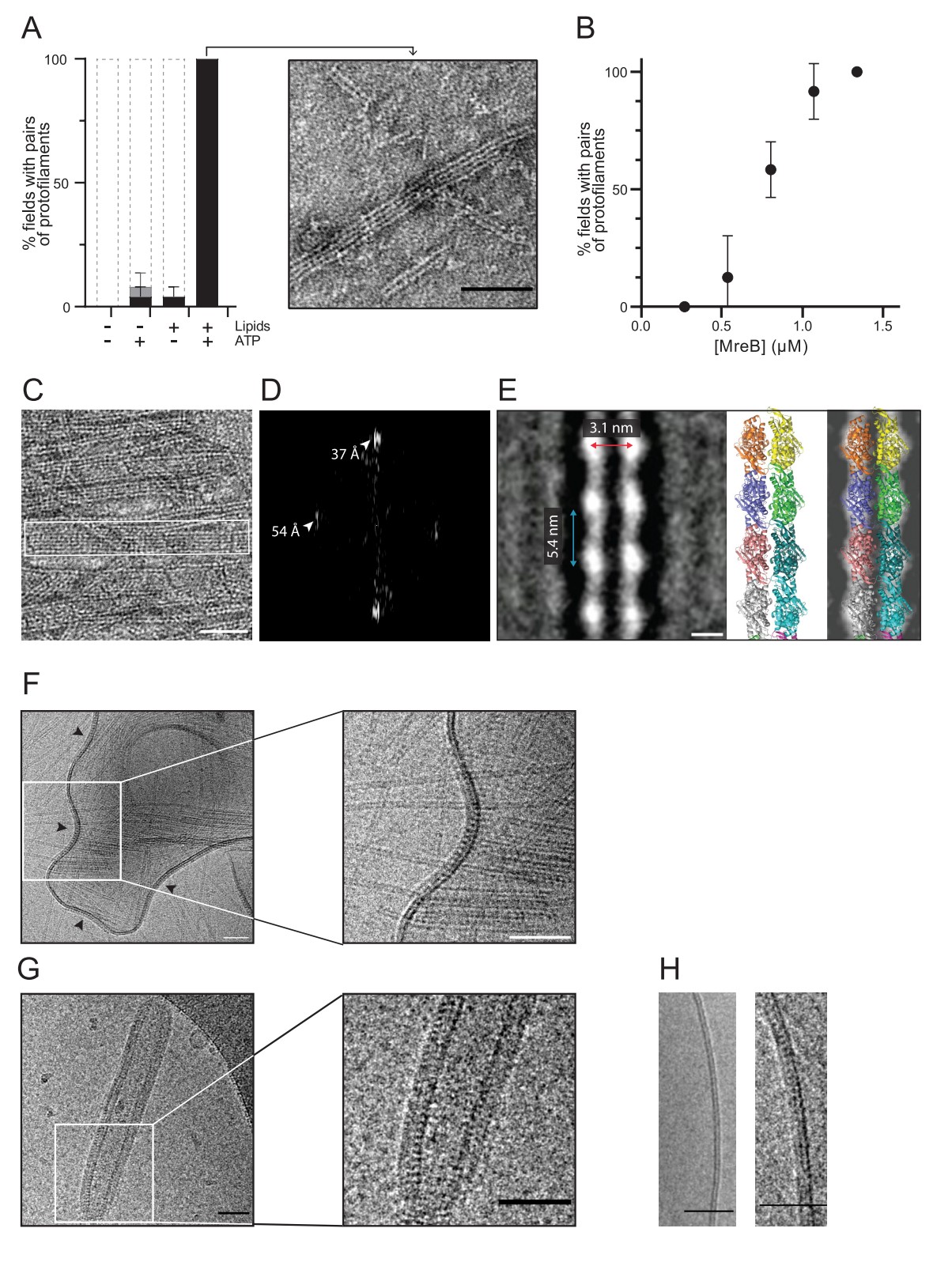

**Figure 2.** MreB[Gs] forms double protofilaments in the presence of ATP and lipids. (**A**) Polymerization of MreB[Gs] into pairs of protofilaments depends on the presence of lipids and ATP. MreB[Gs] was set to polymerize in standard conditions in the presence or absence of ATP and lipid total extract from *E. coli*. Polymer formation is expressed as percent of fields containing high (black) or low (grey) density of polymers (see *Figure 2—figure supplement 1* for details of the quantification method of MreB polymers on TEM grids). Values are average of two independent experiments. Error bars are standard

*Figure 2 continued on next page*

*Figure 2 continued*

deviations. Inset shows an example of a field of dual protofilaments on a negative stained TEM image. Scale bar, 50 nm. (**B**) Polymer formation as a function of MreB^Gs concentration. MreB^Gs was set to polymerize in standard conditions at a concentration ranging from 0.27 to 1.34 μM (0.01–0.05 mg/mL). Values are the average of two independent experiments. Error bars are standard deviations. (**C, D**) MreB^Gs polymers assemble into sheets. (**C**) EM image of MreB^Gs set to polymerize in standard conditions. Scale bar, 50 nm. Fourier transform (**D**) was obtained from the area indicated by a white box in (**C**) and revealed a longitudinal subunit repeat of the filaments of 54 Å and a lateral spacing of ~37 Å (arrowheads). (**E**) (*Left*) 2D averaging of images of negatively stained dual protofilaments of MreB^Gs from 1 554 individual particles. Scale bar, 3 nm. Two copies of the atomic structure of the protofilaments found in the MreB^Gs crystals shown to scale (*Middle*, for illustration the two protofilaments are displayed arbitrarily in an antiparallel conformation but could fit in a parallel conformation as well) and docked into the 2D averaged EM image (*Right*). (**F, G**) MreB^Gs polymers assemble on lipid bilayers and distort liposomes as shown by cryo-electron microscopy (cryo-EM). Cryo-EM micrographs of liposomes (0.37 mg/mL) made from *E. coli* lipid total extracts incubated with purified MreB^Gs (1.34 μM; 0.05 mg/mL) in the presence of ATP (2 mM), and low (F, 100 mM) or high (G, 500 mM) concentration of KCl. Arrowheads point to MreB accumulations. Scale bars, 50 nm. (**H**). Cryo-EM micrographs showing the cross-section of the membrane of liposomes in the absence (*Left*) and in the presence (*Right*) of ATP-bound MreB^Gs at 500 mM KCl. Scale bars, 50 nm.

The online version of this article includes the following figure supplement(s) for figure 2:

**Figure supplement 1.** Workflow for quantification of MreB polymers on TEM grids.

**Figure supplement 2.** MreB presents limited capacity to form polymers in solution.

**Figure supplement 3.** MreB^Gs polymers display a broad range of lengths and widths.

**Figure supplement 4.** 2D averaging of negatively stained images of MreB^Gs dual protofilaments showing the symmetrical arrangement of monomers.

**Figure supplement 5.** MreB^Gs polymers coat and distort liposomes.

hypothesized that the critical concentration for polymerization might be higher in the absence of lipids and thus raised the concentration of MreB^Gs in the reaction (*Figure 2—figure supplement 2B*). Again, virtually no pairs of protofilaments were detected in solution in these conditions. The only polymeric structures observed in the bulk solution, albeit very infrequently and only at high MreB concentration, were some large multilayered sheets forming ribbon-like structures among aggregates (*Figure 2—figure supplement 2B and C*). Taken together, these observations indicated that MreB^Gs polymerization is strongly enhanced by both ATP and lipids.

On a lipid monolayer, polymers were observed at a concentration of MreB above 0.55 μM (0.02 mg/mL), for a theoretical critical concentration of ~0.45 μM (*Figure 2B*), which is very similar to the critical concentration reported for ATP-MreB^Tm (*Bean and Amann, 2008*). The simplest and most abundant assemblies are paired protofilaments (*Figure 2A*, *Figure 2—figure supplements 1 and 3*), as previously observed for MreB^Tm and MreB^Cc assembled on lipid monolayers (*Salje et al., 2011*; *van den Ent et al., 2014*), and for *Spiroplasma* MreBs in solution (*Pande et al., 2022*; *Takahashi et al., 2022*). Pairs of MreB^Gs protofilaments are generally straight, and single protofilaments were never observed. Paired protofilaments of different lengths, ranging from below 50 nm up to several micrometers, as well as partial lateral association into two-dimensional sheets of dual protofilaments are often observed on the same EM grid (*Figure 2A and C*, and *Figure 2—figure supplement 3*). Importantly, pairs of filaments and sheets always lay flat, indicating that they are oriented relative to the membrane surface. The diffraction patterns of the sheets showed a longitudinal repeat of 54 Å and a lateral spacing of ~37 Å (*Figure 2C and D*). 2D averaging of negatively stained EM images of 1 554 individual pairs of filaments (*Figure 2E* and *Figure 2—figure supplement 4*) confirmed a longitudinal subunit repeat of 54 Å and refined the lateral subunit repeat to 31 Å, and could accommodate well two scaled protofilaments found in the MreB^Gs crystals (*Figure 2E*). However, it is not possible to derive the orientation of the two protofilaments (i.e. parallel or antiparallel) from the EM density obtained by 2D averaging.

## Cations modulate distortion of liposomes by MreB^Gs filaments

MreB^Gs filaments also formed on lipid bilayers as observed by cryo-electron microscopy (cryo-EM). To this end, we prepared large unilamellar vesicles (LUVs) from *E. coli* lipid total extract, and incubated them with MreB^Gs and ATP. LUVs alone were spherical (*Figure 2—figure supplement 5A*), but vesicles decorated with MreB^Gs filaments appeared strongly deformed, confirming that MreB^Gs was bound to the membrane. At 100 mM KCl (our standard polymerization condition), LUVs displayed inward bending (negatively curved areas) where MreB^Gs filaments accumulated (arrows in *Figure 2F* and *Figure 2—figure supplement 5B*), as previously reported for MreB^Tm and MreB^Cc (*Salje et al., 2011*; *van den Ent et al., 2014*). 100 mM KCl is a salt concentration commonly used in polymerization studies

of actin and actin-like proteins, including MreB (*Deng et al., 2016*; *Garner et al., 2004*; *Polka et al., 2009*; *Rivera et al., 2011*). Yet, while cytoplasmic $K^+$ concentrations are around 50–250 mM in multicellular eukaryotic cells (*Rodríguez-Navarro, 2000*; *Schmidt-Nielsen, 1975*), they reach 200–300 mM in yeast (*Ariño et al., 2010*) and vary greatly depending on the osmolality of the medium in bacteria (*Cayley et al., 1991*; *Epstein and Schultz, 1965*; *Rhoads et al., 1976*). In *B. subtilis*, the basal intracellular concentration of KCl fluctuates between 100 mM and 800 mM (*Eisenstadt, 1972*; *Whatmore et al., 1990*). We therefore tested how higher salt concentrations affect the properties of MreB polymers. At 500 mM KCl, MreB$^{Gs}$ readily polymerized into straight pairs of filaments as well, which also distorted liposomes but did not induce negative curvature. Instead, they faceted and tubulated the liposomes (*Figure 2G* and *Figure 2—figure supplement 5C*), suggesting that high salt concentration increases the stiffness of MreB filaments and/or of the membrane. Specific binding of cations at discrete sites along the filament has been shown to stiffen actin filaments, determining their bending rigidity (*Kang et al., 2012*). Our result suggests that physiological salt concentrations may also play a fundamental role in the mechanical properties of MreB filaments. MreB$^{Gs}$ largely coated the liposomes and displayed a regular pattern along the cross-section of tubulated vesicles (*Figure 2G–H*). This view is compatible with longitudinal sections of 2D-sheets of straight filaments aligned in parallel to the longitudinal axis of the cylinder, as previously suggested for the arrangement of MreB$^{Tm}$ in rigid lipid tubes (*van den Ent et al., 2014*).

## ATP or GTP drive efficient formation of double filaments on a membrane surface

The role of nucleotide binding and hydrolysis in MreB polymerization remains unclear. In actin, ATP binding or hydrolysis are not required for polymerization (*De La Cruz et al., 2000*; *Kasai et al., 1965*). ATP hydrolysis only occurs subsequent to the polymerization reaction, destabilizing the filaments upon release of the γ-phosphate (*Korn, 1982*; *Korn et al., 1987*). In contrast, MreB$^{Tm}$ was reported to require either ATP or GTP to polymerize (*Esue et al., 2006*; *Nurse and Marians, 2013*; *van den Ent et al., 2001*). MreB from *E. coli*, *C. crescentus*, *S. citri*, and *Leptospira interrogans* also formed polymers in the presence of ATP, but the requirement of ATP for polymerization was not established (*Barkó et al., 2016*; *Harne et al., 2020*; *Maeda et al., 2012*; *Nurse and Marians, 2013*; *Salje et al., 2011*; *van den Ent et al., 2014*). However, MreB filaments or sheets of filaments were also observed in the presence of ADP (*Gaballah et al., 2011*; *Pande et al., 2022*; *Popp et al., 2010b*; *Takahashi et al., 2022*) or AMP-PNP (*Pande et al., 2022*; *Salje et al., 2011*; *Takahashi et al., 2022*). These observations indicated that ATP binding and hydrolysis is not strictly required for filament formation *in vitro*. An analysis of nucleotide-bound crystal structures of MreB$^{Cc}$ also suggested that ATP binding may trigger the transition to the double-protofilament conformation (*Pande et al., 2022*). Furthermore, liposome binding studies of MreB5$^{Sc}$ pointed to an allosteric effect of ATP binding and hydrolysis for effective polymerization and membrane binding (*Pande et al., 2022*).

We then wondered about the specificity of MreB$^{Gs}$ toward nucleotides and their role in polymerization on a lipid membrane. MreB$^{Gs}$ formed straight pairs of protofilaments and sheets in the presence of either ATP or GTP, as shown by negative stain EM (*Figure 3A*). Noteworthy, the average length of double filaments displayed an approximately twofold increase in the presence of GTP compared to ATP (*Figure 3—figure supplement 1A*). The significance of this observation is unclear at present but it may reflect differential affinity, dissociation rate or hydrolytic activity of the two nucleotide triphosphates (NTPs). Next, we asked whether formation of pairs of filaments required nucleotide hydrolysis and tested if nucleotides diphosphate or non-hydrolysable ATP analogues would also support polymer assembly. Virtually no double filaments were observed when ATP/GTP was replaced by ADP, GDP, AMP-PNP, or ApCpp (5′-adenylyl methylenediphosphate), either in the presence or in the absence of lipids, in our standard polymerization conditions (*Figure 3A* and *Figure 3—figure supplement 1B*). However, differential affinity of MreB$^{Gs}$ for these nucleotides, or a higher critical concentration for MreB$^{Gs}$ to polymerize in their presence could also explain these results. Both actin (*Cooke and Murdoch, 1973*; *Iyengar and Weber, 1964*; *Kinosian et al., 1993*) and MreB$^{Cc}$ (*van den Ent et al., 2014*) have the highest affinity for ATP, followed by ADP and then by AMP-PNP. Similarly, the critical concentration of actin polymerized with ADP is about 18-fold higher than with ATP (*Fujiwara et al., 2007*; *Pollard, 1986*), while the critical concentration of *Thermotoga* and *Spiroplasma* ADP-MreBs was reported to be approximately threefold and twofold higher, respectively, than that of

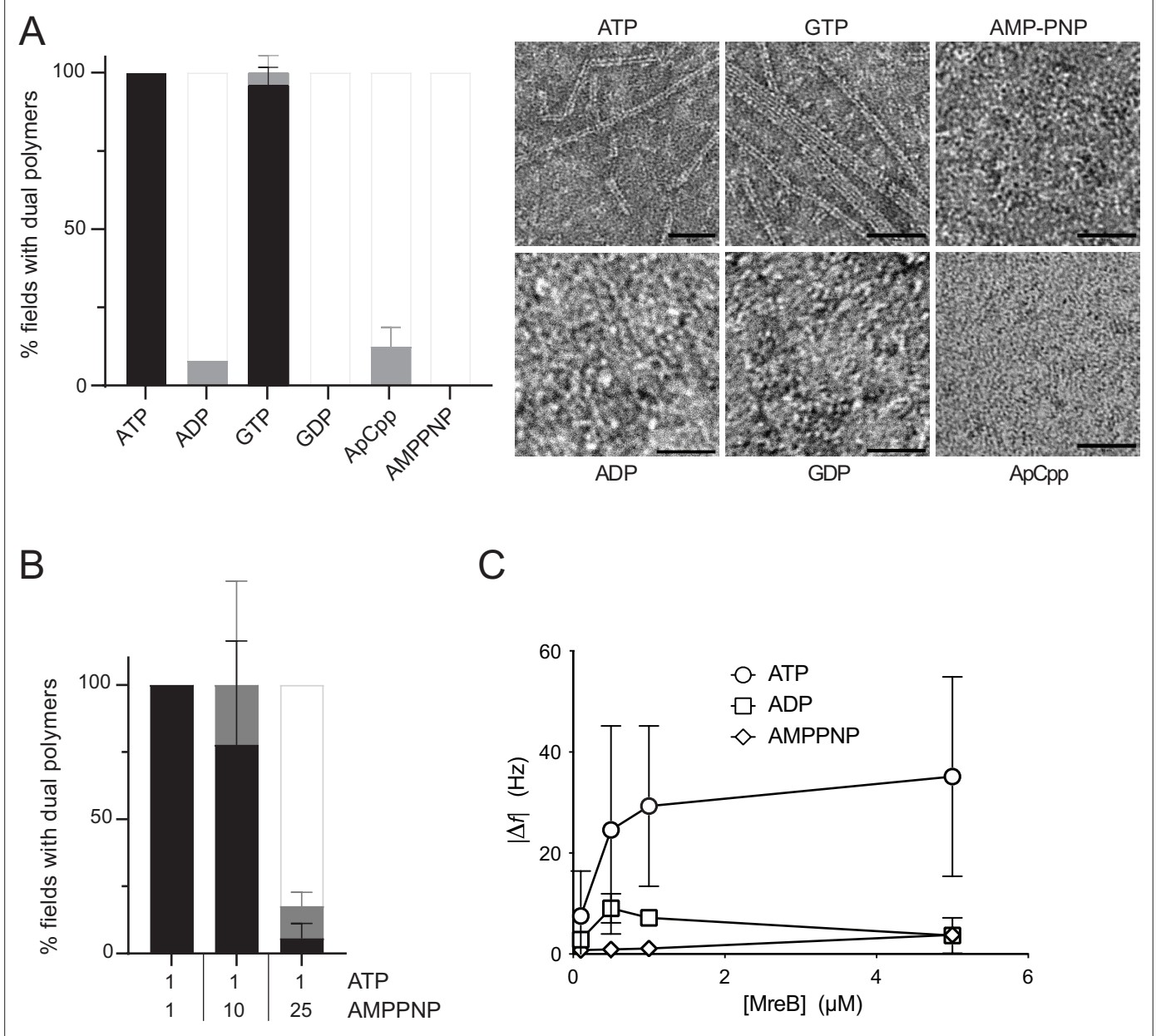

**Figure 3.** Double protofilaments of *Geobacillus* MreB efficiently form in the presence of hydrolysable nucleotides. (**A**) ATP and GTP promote efficient assembly of MreB[Gs] polymers on a lipid surface. MreB (1.34 μM; 0,05 mg/mL) was incubated in the presence of either ATP, ADP, GTP, GDP, or the non-hydrolysable AMP-PNP or ApCpp (2 mM), on a lipid monolayer. (*Left*) Quantification of MreB[Gs] pairs of filaments. Polymer formation is expressed as percent of fields containing high (black), low (grey) density of polymers, or no polymers. Values are averages of at least two independent experiments. Error bars are standard deviations. (*Right*) representative TEM images. Scale bars, 50 nm. (**B**) AMP-PNP-MreB[Gs] does not form double filaments on a lipid monolayer. AMP-PNP competes with ATP for binding to MreB[Gs], preventing polymerization. MreB[Gs] was set to polymerize in standard conditions except that 2 mM ATP was replaced by a mix of ATP and AMP-PNP at the indicated concentrations (in mM). Polymer formation was quantified as in (**A**). Values are average of three independent experiments. Error bars are standard deviations. (**C**) Adsorption of MreB[Gs] to a supported lipid bilayer (SLB) requires ATP. Frequency changes (|Δ*f*|) in QCM-D experiments measured with varying amount (0.1–5 μM) of MreB[Gs] on SLBs made of DOPC:DOPG 80:20 and in the presence of 2 mM of either ATP, ADP, or AMP-PNP. Values are an average of four independent experiments.

The online version of this article includes the following figure supplement(s) for figure 3:

**Figure supplement 1.** Effect of nucleotides, lipids and protein concentration on the polymerization of MreB.

**Figure supplement 2.** Sheet-like structures formed by MreB in solution.

**Figure supplement 3.** QCM-D experiments of MreB[Gs] adsorption on supported lipid bilayers.

ATP-MreBs (*Bean and Amann, 2008*; *Takahashi et al., 2022*). In contrast, the critical concentration of *B. subtilis* MreB deduced from light scattering experiments was 0.9 µM regardless of the nucleotide species bound, and even in the absence of nucleotide (*Mayer and Amann, 2009*). To exclude that the absence of polymerization was due to reduced nucleotide binding, we first increased the concentration of ADP and AMP-PNP from 2 mM to 50 mM. Again, no polymers were detected in the negatively stained samples (*Figure 3—figure supplement 1C*). Next, we performed a competition experiment by mixing ATP (1 mM) with increasing amounts of AMP-PNP (1, 10 and 25 mM) in the polymerization reaction. Increasing amounts of AMP-PNP efficiently decreased the presence of MreB^Gs filaments on the EM grids (*Figure 3B*). We concluded that AMP-PNP binds to MreB^Gs but does not support efficient polymerization on a lipid surface. Next, we tested if the critical concentration of MreB^Gs could be higher in the presence of ADP or AMP-PNP than in the presence of ATP. At the highest MreB^Gs concentration that we could test in our experimental conditions (6.98 µM), about 15-fold the estimated ATP-MreB^Gs critical concentration (0.45 µM, *Figure 2B*), efficient formation of double filaments on the lipid monolayer was still observed only in the presence of ATP (*Figure 3—figure supplement 1D*). Although still scarce, ADP-MreB^Gs double filaments were nevertheless more frequent at higher MreB concentration (*Figure 3—figure supplement 1D*), indicating a higher critical concentration required for filament formation.

Taken together, these results show a noticeable dependence of MreB^Gs filaments on nucleotides and suggest that ATP/GTP binding and/or hydrolysis is required for efficient assembly of MreB^Gs into pairs of filaments on a membrane surface.

## Nucleotide hydrolysis mediates binding of MreB^Gs to the membrane

Surprisingly, at high MreB concentrations, the large sheet- and ribbon-like structures observed to form in the bulk solution at very low frequency in the presence of ATP (*Figure 2—figure supplement 2B and C*) became frequent in the presence of ADP and very frequent in the presence of AMP-PNP (*Figure 3—figure supplement 2A, B*). The Fourier transforms of the sheet-like structures formed in these conditions showed sets of discrete reflections, with diffraction equatorial spots of about 6.5 nm (*Figure 3—figure supplement 2C*), larger than the repeats found on membrane-bound sheets, suggesting that membrane binding induces conformational changes in MreB filaments, as previously suggested (*Shi et al., 2020*). The presence of ADP- and AMP-PNP-MreB^Gs sheets in solution indicated that nucleotide hydrolysis is not strictly required for filament formation. However, the formation of extended sheets in solution concomitant with the absence of double protofilaments on a lipid surface in the presence of ADP or AMP-PNP, suggests that ATP hydrolysis promotes efficient filament formation at the membrane.

We then wondered whether ATP hydrolysis mediates MreB^Gs membrane binding or whether it triggers polymerization of membrane-bound monomers. To address this question, we turned to quartz crystal microbalance with dissipation monitoring (QCM-D) to measure the binding affinity of MreB^Gs to supported lipid bilayers (SLBs) of various lipid mixtures. QCM-D is a surface-sensitive technique that can be used to measure biomolecular interactions at aqueous interfaces in real time (*Reviakine et al., 2011*). Changes in frequency ($\Delta f$) and dissipation ($\Delta D$) are recorded. The frequency is directly proportional to any mass added or removed (*Sauerbrey, 1959*), while dissipation changes are indicative of the viscoelastic properties of the attached layer. QCM-D was previously applied to study, for example, the binding affinity of the division proteins MinD and MinE of *E. coli* to SLBs (*Renner and Weibel, 2012*). *E. coli* and *B. subtilis* cytoplasmic membranes are mainly composed of phospholipids, with the negatively charged phosphatidylglycerol (PG) and the zwitterionic phosphatidylethanolamine (PE) being the dominant species (*Bernat et al., 2016*; *Bishop et al., 1967*; *den Kamp et al., 1969*; *Laydevant et al., 2022*; *Nickels et al., 2017*; *Seydlová and Svobodová, 2008*; *Sohlenkamp and Geiger, 2016*). Although lipid proportions vary widely depending on the strains and growth conditions, PE is largely dominant in *E. coli* while PG is more dominant in *B. subtilis*, indicating that phospholipids are more negatively charged in Gram-positive membranes. To mimic *Bacillus* membranes in our QCM-D assay, we used mixtures of the zwitterionic dioleoylphosphatidylcholine (DOPC) doped with the anionic dioleoylphosphatidylglycerol (DOPG) in various ratios (DOPC:DOPG 100:0, 90:10 or 80:20) to generate SLBs. DOPC was selected to replace PE because of its widespread role as a scaffold lipid in SLBs formation. We had to adopt a mixture that enabled us to form SLBs on planar substrates, as the inverted conical shape of PE makes the formation of planar SLBs difficult (PE tends

to form non-bilayer structures because of its small headgroup). A typical SLBs signature experiment is shown in *Figure 3—figure supplement 3A–B*. Briefly, SLBs are formed after the adsorption of liposomes ($\Delta f$ decrease, $\Delta D$ increase) onto activated silica surfaces. Once a critical surface concentration of liposomes is reached and the interactions between liposomes and the surface are suitable, the liposomes spontaneously rupture and coalesce into flat SLBs (*Keller et al., 2000*). After the formation of stable and flat SLBs (i.e. a stable baseline for frequency and dissipation; *Figure 3—figure supplement 3A*), we started to add MreB[Gs] to the SLBs (*Figure 3—figure supplement 3B*, closed arrows). We recorded frequency and dissipation changes for the added MreB[Gs] protein in varying concentrations on all SLBs. Binding was strongly dependent on ATP and was substantially affected by the lipid composition of SLBs (*Figure 3C* and *Figure 3—figure supplement 3C and D*). Increasing the levels of DOPG led to a higher amount of MreB[Gs] binding, with DOPC:PG 80:20 giving the highest observed adsorption, suggesting that the presence of negatively charged lipids favors MreB[Gs] binding to the membrane. In the presence of ATP, binding was detected almost instantaneously after adding MreB[Gs] (*Figure 3—figure supplement 3B*, closed arrows) for all concentrations of MreB tested, either above or below the concentration in which polymers were observed by EM on a lipid monolayer (0.55 µM) (*Figure 2B* and *Supplementary file 2*). Interestingly, a shift in frequency was detected even at 0.1 µM MreB, well below the critical ATP-MreB concentration, suggesting that MreB monomers might bind too (*Figure 3C* and *Figure 3—figure supplement 3B and E, F*). The protein binding kinetics reached an equilibrium after approximately 5–10 min with a somewhat slower continued binding of additional MreB[Gs] molecules (*Figure 3—figure supplement 3B*). Upon rinsing with the same buffer (*Figure 3—figure supplement 3B*, open arrows), MreB[Gs] at low concentration (0.1 µM) was almost completely removed from the membrane while at higher concentration (1 µM) it remained stably absorbed. When replacing ATP with ADP or AMP-PNP, we were not able to detect any significant binding at either low or high concentrations of MreB, indicating a virtually complete loss of interaction (*Figure 3C* and *Figure 3—figure supplement 3D and F*). We further increased the concentration of ADP or AMP-PNP to exclude the possibility that the binding was simply affected by a decreased affinity of MreB[Gs] for these nucleotides. Higher concentrations of ADP and AMP-PNP did not restore the binding of MreB[Gs] to the SLBs (*Figure 3—figure supplement 3G*). Taken together, these results suggest that nucleotide hydrolysis drives MreB[Gs] membrane binding. Because some binding was detected at 0.1 µM MreB in the presence of ATP but not of ADP and AMP-PNP, it is tempting to speculate that lipid binding might occur prior to polymerization into pairs of filaments on the membrane surface.

Finally, we used the Sauerbrey model (*Sauerbrey, 1959*) to calculate the average coverage and thickness of the layer of MreB[Gs] attached to the SLB. The thickness of the MreB films ranged from 0.1 nm to approximately 4 nm on the SLBs with an 80:20 ratio of DOPC:DOPG, which corresponds to ~2.5% to 100% coverage assuming a monolayer filament thickness (*Figure 3—figure supplement 3H* and Materials and methods). These data suggest that MreB[Gs] mainly form monolayers on the SLBs, with limited out-of-plane interactions (i.e. limited tendency to stack into multilayers), consistent with our EM observations of pairs of filaments and sheets lying flat on the lipid monolayer (*Figure 2*, *Figure 3* and *Figure 2—figure supplements 1 and 4*) and with the pattern displayed by the filaments on cross-sections of vesicles (*Figure 2G*), and thus with the interaction of the membrane with a specific surface of the MreB[Gs] filaments. Taken together, these observations suggest an oriented arrangement of MreB[Gs] filaments on the membrane, with lateral interactions between filaments in the plane perpendicular to their membrane-binding surface.

## The hydrophobic amino-terminus and α2-β7 loop of MreB[Gs] are required for membrane anchoring alongside with electrostatic interactions

Membrane binding of MreB[Tm] is mediated by a small loop containing two hydrophobic residues (L93 and F94), whereas binding of MreB[Ec] and MreB[Cc] is mediated by an amino-terminal extension (9 residues) predicted as an amphipathic helix, which is disordered in all crystal structures of MreB[Cc] (*Salje et al., 2011*; *van den Ent et al., 2014*; *Figure 1—figure supplement 1*, green highlights). Albeit essential to MreB function in *E. coli* (*Salje et al., 2011*), this N-terminal extension is not required for polymerization *in vitro* (*Salje et al., 2011*; *van den Ent et al., 2014*). MreB[Bs] was not predicted to carry an N-terminal amphipathic helix (*Salje et al., 2011*). A systematic search in a large panel of MreB proteins spanning over the entire bacterial kingdom revealed that N-terminal amphipathic helices are a conserved feature of the Proteobacteria phylum and most Gram-negative bacteria, but are absent

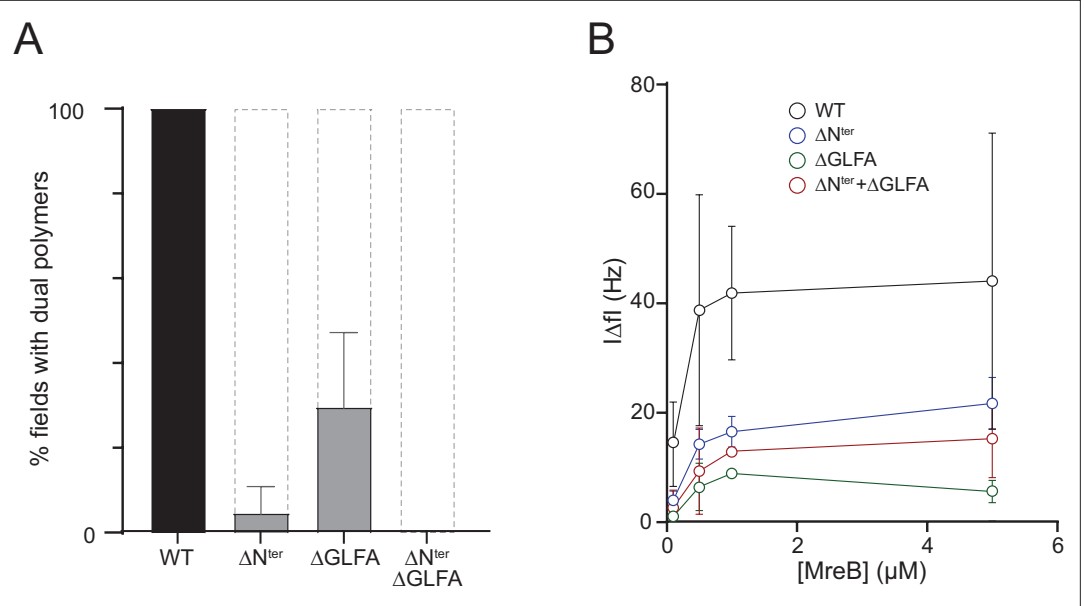

**Figure 4.** in theThe N-terminus and the α2β7 hydrophobic loop of MreB<sup>Gs</sup> promote membrane binding and polymerization on a lipid surface. (**A**) Both the hydrophobic α2-β7 loop and the N-terminus sequence of MreB<sup>Gs</sup> are required for efficient polymerization on a lipid monolayer. Frequency and density of polymer formation in high salt (500 mM KCl) polymerization conditions, observed on negatively stained TEM images for the wild type (WT) and the mutants of the α2-β7 loop (ΔGLFA), the N-terminus (ΔN<sup>ter</sup>) or both domains (ΔN<sup>ter</sup>+ ΔGLFA) of MreB<sup>Gs</sup>. Images were categorized based on the absence or the presence of low or high density of polymers. Values are the average of two independent experiments. Error bars are standard deviations. (**B**) The α2-β7 loop and the N-terminus domains of MreB<sup>Gs</sup> enhance its adsorption to supported lipid bilayers. Frequency change (IΔfl) measured for the binding of various concentrations (0.1–5 µM) of purified wild-type (WT) and mutant forms of MreB<sup>Gs</sup> to SLBs. Incubations were performed in polymerization buffer containing 500 mM KCl and 2 mM ATP. SLBs contained an 80:20 molecular ratio of DOPC:DOPG. Values are an average of at least two independent experiments.

The online version of this article includes the following figure supplement(s) for figure 4:

**Figure supplement 1.** Distribution of N-terminal amphipathic helices and hydrophobic sequences on MreBs proteins in the bacterial kingdom.

**Figure supplement 2.** The protruding hydrophobic subdomain IA of MreB<sup>Gs</sup> is surrounded by a positively charged cluster.

**Figure supplement 3.** Circular dichroism (CD) spectra showing similar folding of the wild-type and the deletion mutants of the α2-β7 loop (ΔGLFA), the N-terminus (ΔN<sup>ter</sup>) or both domains (ΔN<sup>ter</sup> + ΔGLFA) of recombinant MreB<sup>Gs</sup>.

**Figure supplement 4.** The amino-terminal sequence, the GLFA residues of the α2-β7 hydrophobic loop and electrostatic interactions mediate binding of MreB to a lipid surface.

**Figure supplement 5.** Crystal structure of MreB<sup>Gs</sup> bound to ATP.

in *Firmicutes* and *Bacteroidetes* species (*Figure 4—figure supplement 1*). Most *Firmicutes*, including *Bacilli* (MreB<sup>Gs</sup> and MreB<sup>Bs</sup>) and *Clostridia*, but with the notable exception of the wall-less *Mollicutes*, possess a shorter N-terminal sequence containing 4–7 hydrophobic amino-acids (*Figure 1—figure supplement 1* and *Figure 4—figure supplement 1*). We noticed that in the crystal structure of the apo form of MreB<sup>Gs</sup> this short hydrophobic N-terminal sequence is in close proximity to loop α2-β7 (*Figure 1A*), which in MreB<sup>Tm</sup> carries the hydrophobic residues L93 and F94 involved in membrane binding (*Salje et al., 2011*). The α2-β7 loops of MreB<sup>Bs</sup> and MreB<sup>Gs</sup> contain additional hydrophobic residues (*Figure 1—figure supplement 1*), suggesting that they may also play a role in membrane interaction. Interestingly, analysis of the crystal structure of MreB<sup>Gs</sup> also showed that a cluster of positively charged residues surrounds the protruding hydrophobic subdomain formed by the α2-β7 loop and the N-terminal sequence (*Figure 4—figure supplement 2*). Since our QCM-D data suggested that anionic lipids favor MreB<sup>Gs</sup> binding to a lipid bilayer, we hypothesized that membrane binding

may result from both ionic and hydrophobic interactions, with the positively charged residues that cluster around the membrane insertion hydrophobic region interacting with negatively charged lipids.

We then constructed and purified mutants deleted for either four hydrophobic residues of the α2-β7 loop (aa 95–98, GLFA), the N-terminal sequence (aa 2–7, FGIGTK), or both (*Supplementary file 3*). Folding of the protein was not affected by the deletions as shown by circular dichroism (CD) (*Figure 4—figure supplement 3*). To comparatively analyze membrane binding of these mutants, we first examined their polymerization at high-salt concentration (500 mM KCl). We reasoned that high salt would reduce the contribution of electrostatic interactions to membrane binding by screening the charges of the anionic lipids in the membrane, allowing to better assess the contribution of the short hydrophobic sequences to membrane anchoring. The three mutants and the wild-type MreB$^{Gs}$ protein were set to polymerize in the presence of lipids and the formation of filaments was assessed by negative stain EM. The three mutants displayed a dramatic reduction in the formation of double filaments in the EM fields relative to the wild-type, with a gradation of defects, with the deletion of the GLFA sequence having the least effect and the double deletion having the greatest (*Figure 4A*). A similar gradient of phenotypes was observed at lower salt concentration (100 mM KCl) but the overall number of filaments detected on the membrane surface was increased as expected, consistent with electrostatic interactions also contributing to lipid binding (*Figure 4—figure supplement 4A*). Under the same polymerization conditions, no filaments were detected in solution for any of the mutants (*Figure 4—figure supplement 4A*), arguing against the possibility that the mutants could form filaments in solution but that they are impaired for membrane binding.

In QCM-D experiments, membrane adsorption in the presence of ATP was strongly reduced in the three mutants relative to the wild-type protein, mirroring the polymerization assays at either high- or low-salt concentration (*Figure 4B* and *Figure 4—figure supplement 4B*). High salt reduced membrane binding relative to low salt in all conditions (*Figure 4—figure supplement 4C*). In the presence of ADP, binding was not observed for any of the mutants, as previously observed with the wild-type protein (*Figure 4—figure supplement 4D*). Taken together, these results show that the spatially close hydrophobic N-terminus and α2-β7 loop are membrane anchors of MreB$^{Gs}$ filaments and further confirm that electrostatic interactions are also involved in lipid binding.

## γ-phosphate dissociation after ATP/GTP hydrolysis by MreB$^{Gs}$ is related to filament turnover

Our results suggest that MreB$^{Gs}$ has an intrinsic affinity for lipids, with nucleotide hydrolysis involved in the switch of the protein from a soluble to a lipid-affine form, potentially through structural modifications. In order to test the impact of nucleotide binding, we co-crystallized MreB$^{Gs}$ with ATP and solved the crystal structure of the complex at 2.3 Å resolution (PDB ID 8AZG). The crystals diffracted in space group P2$_1$2$_1$2 (*Supplementary file 1*) with one molecule per asymmetric unit. The electron density map clearly revealed the presence of three phosphates, demonstrating that the ATP had not been hydrolyzed into ADP (*Figure 4—figure supplement 5A*). This is probably due to the absence of a Mg$^{2+}$ ion in the catalytic site. No residual electron density peak could be interpreted as a Mg$^{2+}$ ion with appropriate coordination. As a consequence, the binding mode of the nucleotide is distinct from other MreB structures, with the γ-phosphate located in the Mg$^{2+}$ binding site (*Figure 4—figure supplement 5B*). The structure of the ATP-bound form of MreB$^{Gs}$ is highly similar to the apo form, with a rmsd of 1.41 Å over 313 aligned Cα atoms (*Figure 4—figure supplement 5C*). However, ATP binding induced a small closure of the nucleotide-binding pocket, and loop β6-α2, which was disordered in the apo structure, is fully visible in the electron density map. The hydrophobic loop α2-β7 and the N-terminus also display an alternative conformation. Despite multiple co-crystallization trials, in the ATP-bound state the crystal packing only revealed monomers and never straight protofilaments as in the apo structure.

MreBs of several Gram-negative bacteria have been shown to slowly hydrolyze ATP in solution (*Bean and Amann, 2008*; *Esue et al., 2005*; *Esue et al., 2006*; *Gaballah et al., 2011*; *Mayer and Amann, 2009*; *Nurse and Marians, 2013*; *Pande et al., 2022*; *Popp et al., 2010b*). Our EM and QCM-D results suggested that ATP hydrolysis by MreB$^{Gs}$ may be required for efficient membrane binding and polymerization, and thus that hydrolysis may occur in solution also. We monitored ATPase activity by measuring the release of inorganic phosphate (P$_i$) in the presence of ATP for a wide range of MreB concentrations, in the presence and in the absence of lipids. In our standard polymerization

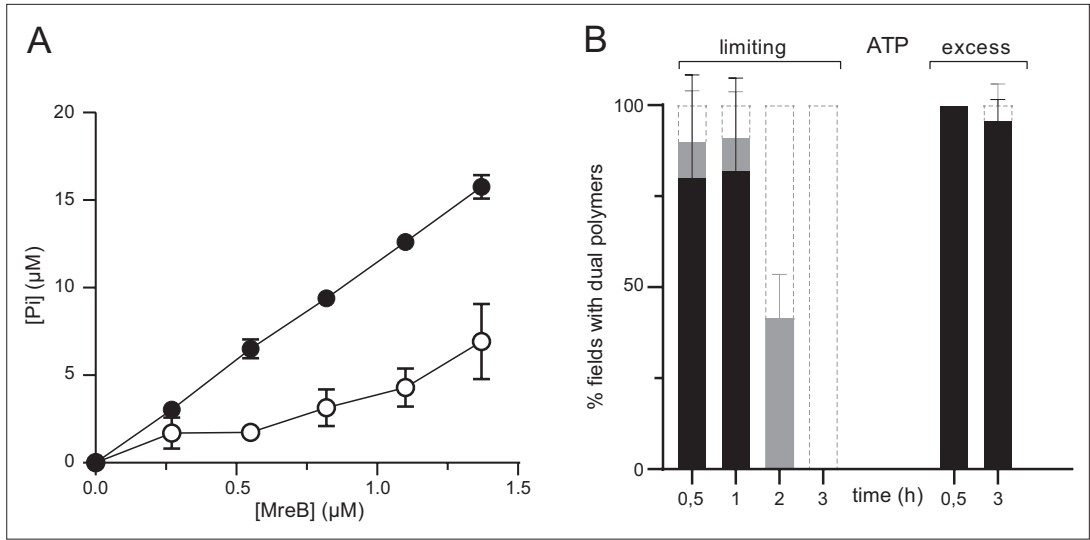

**Figure 5.** ATPase activity of MreB^Gs. (**A**) The ATPase activity of MreB^Gs is stimulated in the presence of lipids. ATPase activity, measured by monitoring inorganic phosphate (Pi) release, of MreB^Gs at different concentrations (0.26–1.34 μM) in the presence of 0.5 mM ATP, 100 mM KCl, and in the presence (filled circles) or absence (empty circles) of 0.05 mg/mL liposomes, after 1 hr incubation at 53 °C. Values are averages of at least two independent experiments. Error bars are standard deviations. (**B**) Kinetics of MreB^Gs polymer formation in the presence of limiting ATP. MreB^Gs was set to polymerize in the presence of standard (2 mM; 'excess') or low (13 μM; 'limiting') concentration of ATP. Samples were prepared for TEM observation at different incubation times, up to 3 hr. Polymer formation is expressed as percent of fields containing high (black) or low (grey) density of polymers, or no polymers. Values are averages of two independent experiments. Error bars are standard deviations.

The online version of this article includes the following figure supplement(s) for figure 5:

**Figure supplement 1.** Effects of lipids, temperature and deletions on the hydrolytic activity of MreB.

conditions, in the absence of lipids and at 53 °C (a temperature close to the optimal growth temperature of *G. stearothermophilus*), we observed a constant release of $P_i$ over time, with an equilibrium rate of $P_i$ dissociation of 0.198±0.008 $P_i$/min/MreB molecule (***Figure 5A***, ***Figure 5—figure supplement 1D***). The rate was reduced approximately threefold, to 0.068±0.02 $P_i$/min/MreB molecule in the absence of lipids (***Figure 5A***, ***Figure 5—figure supplement 1D***). Similar rates were observed when the incubations were performed at a higher KCl concentration (500 mM), with 0.158±0.003 and 0.081±0.004 $P_i$/min/MreB with and without lipids, respectively (***Figure 5—figure supplement 1A-E***). As expected, $P_i$ release rates were decreased when incubations were performed at 37 °C instead of 53 °C, but the rate of $P_i$ release remained comparatively higher in the presence of lipids (***Figure 5—figure supplement 1A***). These rates of $P_i$ release in the presence of ATP (~1 $P_i$/MreB in 5 min at 53 °C) are comparable to those observed for MreB^Tm and MreB^Ec *in vitro* (***Esue et al., 2005***; ***Esue et al., 2006***; ***Nurse and Marians, 2013***), and also remarkably similar to the rate of the (very slow) dissociation of γ-phosphate after ATP hydrolysis within actin filaments, which has a half-time of ~6 min (***Carlier and Pantaloni, 1986***). Interestingly, the release of $P_i$ was constant over the length of our ATPase experiments (***Figure 5—figure supplement 1B***). However, similar density and lengths of negatively stained MreB^Gs polymers were observed over the EM grids for all incubation (polymerization) times tested, ranging from a few minutes to several hours. This suggests that after a fast polymerization step, either (i) filaments remain stable and release $P_i$ constantly but very slowly, or (ii) $P_i$ release reflects the turnover of the filaments. To test these hypotheses, we performed a series of semi-quantitative TEM experiments in the presence of a limiting concentration of ATP. We reasoned that if the population of filaments is in steady-state, polymerization should be unaffected at early time points but polymer density would decrease as ATP gets depleted and, conversely, if the polymers are static, they should not be affected over time. As shown in ***Figure 5B*** a similar density of double protofilaments was observed on the lipid monolayer at early time points at both saturating and limiting concentrations of ATP. However, polymers progressively disappeared over time in the presence of limiting

concentrations of ATP, but not when ATP was in excess. These observations suggest that MreB polymerization on a lipid surface is a dynamic process, with steady-state polymerization/depolymerization rates.

We have shown that MreB[Gs] polymerizes into pairs of protofilaments in the presence of lipids and either ATP or GTP (*Figure 3A* and *Figure 3—figure supplement 1A*). MreB[Tm] was also reported to polymerize in solution in the presence of ATP or GTP (*Bean and Amann, 2008*; *Esue et al., 2006*; *Nurse and Marians, 2013*; *Popp et al., 2010a*; *van den Ent et al., 2001*), and to release $P_i$ at similar rates upon GTP and ATP hydrolysis (*Esue et al., 2006*). We found that MreB[Gs] also releases $P_i$ after hydrolysis of GTP as efficiently as after hydrolysis of ATP, both in the presence and in the absence of lipids (*Figure 5—figure supplement 1C*). Finally, the specific ATPase activity of the three mutants impaired for membrane binding and of the wild-type MreB[Gs] reflected their polymerization and QCM-D assays at either high- or low-salt concentration (*Figure 5—figure supplement 1D*, E). In the absence of lipids, all proteins released $P_i$ at comparative rates, suggesting that a basal level of ATP degradation occurs in solution. In the presence of lipids, the rate of $P_i$ release of all mutants was reduced relative to the wild-type MreB[Gs], and overall $P_i$ release rates were significantly reduced at high salt (500 mM KCl) relative to low salt (100 mM KCl) (*Figure 5—figure supplement 1D–E*). At 100 mM KCl, the mutant deleted for both the GLFA and the N-terminus hydrophobic sequences was the only one severely impacted in $P_i$ release rate (*Figure 5—figure supplement 1D*), as it was in the formation of filaments on a lipid surface (*Figure 4—figure supplement 4A*). We concluded that the release of $P_i$ by the wild-type and membrane-binding mutants in the presence of lipids was consistent with the density of double filaments observed in the EM fields.

Taken together, these results suggest that the presence of lipids is not required for the ATPase/GTPase activity of MreB[Gs]. However, $P_i$ release is enhanced in the presence of lipids, advocating for some conformational changes upon binding of MreB[Gs] to the membrane and/or upon polymerization on the lipid surface. Furthermore, we show that pairs of filaments formed on a lipid surface are at steady-state, undergoing a dynamic, ATP-driven assembly/disassembly process.

## Discussion

Here, we show that bacterial actin MreB from the Gram-positive bacterium *G. stearothermophilus* polymerizes into pairs of protofilaments on lipid surfaces and that this process is dynamic, with steady-state polymerization/depolymerization of the population of filaments. If individual double filaments are antiparallel (*van den Ent et al., 2014*), they are structurally nonpolar and thus their disassembly could result either from dynamic instability or from steady-state fluctuations (growing and shrinking) of the filaments ends. The requirement for a membrane is consistent with the observation that *in vivo* MreB polymeric assemblies are membrane-associated (i.e. localize at the cell periphery), in line with their role as scaffold of the CW elongation machinery. Membrane binding of MreB[Gs] is direct and mediated by the hydrophobic α2-β7 loop protruding from the protein in domain IA, consistent with the prediction by Salje and colleagues that binding to membranes via such a hydrophobic loop and/or an amphipathic helix may be conserved for all MreBs (*Salje et al., 2011*). However, we found that MreB[Gs] membrane binding is also mediated by the hydrophobic N-terminus, which together with the spatially closed α2-β7 loop would constitute a membrane anchor, as well as by electrostatic interactions, possibly through the positively charged residues around the membrane insertion hydrophobic region. The absence of an amphipathic helix and the presence instead of a hydrophobic N-terminus in many MreB sequences (*Figure 4—figure supplement 1*) suggest that most MreB use one or the other amino-terminal structure to bind to membranes.

Another important finding concerns the role of NTP in MreB[Gs] membrane binding and formation of double protofilaments on a lipid surface. In the absence of lipids, MreBs from Gram-negative bacteria have been reported to assemble into sheets and bundles in the presence of either ATP or non-hydrolysable nucleotides (*Barkó et al., 2016*; *Esue et al., 2005*; *Esue et al., 2006*; *Harne et al., 2020*; *Maeda et al., 2012*; *Nurse and Marians, 2013*; *Popp et al., 2010b*; *Salje et al., 2011*; *van den Ent et al., 2001*), as we found here for MreB[Gs], indicating that membrane binding and nucleotide hydrolysis are not a prerequisite for polymerization. However, our data indicate that ATP binding and hydrolysis promote effective membrane binding and transition to the double-protofilament conformation, as recently suggested for *Spiroplasma* MreB5 (*Pande et al., 2022*). Efficient formation of double filaments on the lipid monolayer was not observed in the presence of ADP, even at the

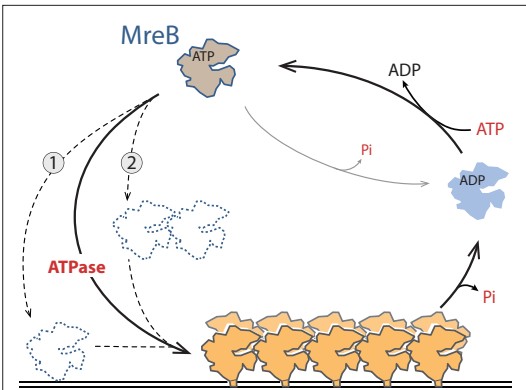

**Figure 6.** Model for ATP-driven MreB$^{Gs}$ membrane binding and polymerization into pairs of filaments. ATP hydrolysis stimulates MreB$^{Gs}$ adsorption to lipids, possibly by promoting a conformational change that renders the hydrophobic α2-β7 loop and N-terminal protruding region prone for insertion into the membrane, either as a monomeric (1) or a nucleated (2) MreB$^{Gs}$ form (dotted lines). Lipid binding would trigger formation of double-protofilaments on the lipid surface, which in turn would promote Pi release. Membrane-associated pairs of filaments may be mostly in the ADP-P$_i$ form, and Pi release may destabilize them and ultimately promote disassembly.

highest MreB$^{Gs}$ concentration that we could test, about 15-fold the estimated ATP-MreB$^{Gs}$ critical concentration. The critical concentration of actin polymerized with ADP is 18-fold higher than that of actin polymerized with ATP (*Fujiwara et al., 2007*; *Pollard, 1986*). This suggests that depolymerisation of the membrane-bound double filaments is stimulated by P$_i$ release, as F-actin depolymerisation.

Taken together, our data suggest a model for ATP-driven polymerization of MreB$^{Gs}$ on a lipid surface (*Figure 6*). Comparison of the crystal structures of apo and ATP-bound MreB$^{Gs}$ shows that only minor conformational changes occur upon nucleotide-binding, in agreement with what was observed when comparing crystal structures of MreB$^{Cc}$ and MreB5$^{Sc}$ in different nucleotide-bound states (*Harne et al., 2020*; *Pande et al., 2022*; *van den Ent et al., 2014*). This invariability of folding regardless of the bound ligands has also been observed in crystal structures of actin and other members of the actin superfamily (*Schüler, 2001*). ATP hydrolysis might promote small but dynamic structural changes that cannot be observed in crystal structures, which are locked in a conformation imposed by the packing. Different modes of membrane binding depending on the nucleotide state have indeed been suggested for

MreB based on molecular dynamics simulations (*Shi et al., 2020*). It is tempting to speculate that MreB has limited affinity for lipids, and nucleotide hydrolysis may induce a conformational change that allows or stabilizes its interaction with the membrane. This is consistent with our findings that ATP is required for membrane interaction in QCM-D experiments and for detection of pairs of filaments on a lipid surface, and that at high concentration of MreB, ADP- and AMP-PNP-MreB sheets are frequent in solution while pairs of filaments found on the membrane are, respectively, scarce and virtually absent. Membrane interaction could occur by either a monomeric or an oligomeric form of MreB (*Figure 6*). It was previously speculated that the weak binding energy of individual membrane binding domains could be overcome by multimerization (*Salje et al., 2011*), which would favor the latter. Such oligomerization would nonetheless remain limited to very short filaments, as a nucleation step, since no filaments were detected in solution in these conditions. Binding to lipids would induce a second conformational change that triggers polymerization into double protofilaments (*Figure 6*). The double filament conformation might in turn favor P$_i$ release, which would ultimately promote filament disassembly. In this scenario, the ADP-P$_i$-MreB intermediate would be the long-lived intermediate state within MreB filaments. The rate of P$_i$ release from MreB$^{Gs}$ is consistent with previous reports on MreB$^{Tm}$, MreB$^{Ec}$ and MreB5$^{Sc}$ (*Bean and Amann, 2008*; *Esue et al., 2005*; *Esue et al., 2006*; *Nurse and Marians, 2013*) and is strikingly similar to that from F-actin. It will be interesting to measure the half-time of ATP hydrolysis by MreB$^{Gs}$ and confirm if it is uncoupled and much faster than the half-time of P$_i$ release, like in F-actin (~2 s and ~6 min, respectively; *Pollard, 2016*). In the absence of lipids, the decreased P$_i$ release rate observed may reflect either a reduced P$_i$ dissociation from soluble MreB-ADP-P$_i$ or a reduced ATPase activity of the protein.

Our cryo-EM data show that MreB$^{Gs}$ filaments are straight and rigid at high KCl concentration, strongly deforming and faceting lipid vesicles, whereas at lower salt concentration, they bend the liposomes into negatively curved vesicles as previously reported for MreB$^{Tm}$ and MreB$^{Cc}$ (*Salje et al., 2011*; *van den Ent et al., 2014*; *Figure 2F–G* and *Figure 2—figure supplement 5*). A recent model postulates that MreB polymers are intrinsically curved and have a strong affinity for negatively curved membranes while avoiding to be positively bent (*Hussain et al., 2018*; *Wong et al., 2019*). The

presence of a high concentration of KCl could stiffen MreB filaments, as shown to occur for actin filaments (**Kang et al., 2012**), and/or the lipid membrane, leading to a virtual disappearance of curved liposomes *in vitro* in these conditions. Because in bacterial cells the intracellular KCl concentration varies, ranging from 100 mM to 800 mM in *B. subtilis* (**Eisenstadt, 1972**; **Whatmore et al., 1990**), it questions the ability of MreB to induce or sense membrane curvature *in vivo*. In any case, the intrinsic affinity of MreB protofilaments for negative concave membrane curvature remains to be demonstrated. The identity of the bound nucleotide may also regulate the bending properties of MreB filaments, as it has been shown for actin filaments, with ADP-bound actin filaments being less stiff than ATP-bound filaments (**Isambert et al., 1995**). In agreement with this, molecular simulations suggest that MreB filaments exhibit nucleotide-dependent intersubunit bending, yet with straight, ADP-bound and curved, ATP-bound filaments (**Colavin et al., 2014**), opposite to actin. The kinetics of polymerization of MreB filaments, the apolarity of growth of presumably antiparallel doublets, and whether their disassembly results from dynamic instability or from growing and shrinking of the filaments at steady-state as well as the characterization of the polyelectrolyte properties of MreB filaments and their interactions with lipids are also important questions for future studies.

## Materials and methods

### General procedures and growth conditions

DNA manipulations were carried out by standard methods. *G. stearothermophilus* was grown at 55 °C and *E. coli* at 37 °C in rich lysogeny broth (LB). Kanamycin was used at 25 µg/mL. All strains used in this study are listed in **Supplementary file 4**. All lipids, *E. coli* lipid Total Extract (TE), 1,2-Dioleoyl-sn-glycero-3-phosphocholine (DOPC), and 1,2-dioleoyl-sn-glycero-3-phospho-(1′-rac-glycerol) (DOPG), were purchased from Avanti Polar Lipids Inc (Alabaster, AL, USA).

### Cloning, expression and purification of MreB variants from *G. stearothermophilus*

Two *mreB* paralogs were identified in the genome of *G. stearothermophilus* ATCC 7953, corresponding to *mreB* and *mbl* of *B. subtilis* based on their synteny. The *mreB* ortholog displays a strong 92.6% similarity (85.6% overall identity) with *mreB* of *B. subtilis* (**Figure 1—figure supplement 1**). *mreB* from *G. stearothermophilus* ATCC 7953 was amplified by PCR using primers cc430 and cc431 (**Supplementary file 5**) and *G. stearothermophilus* growing cells as template. A second DNA fragment was generated by PCR on a derivative of plasmid pET28a (devoid of the first three codons following the *Nco*I restriction site), using primers cc433/cc432 (**Supplementary file 5**). The two resulting fragments were assembled by isothermal assembly and transformed into *E. coli* (**Gibson et al., 2009**). The resulting plasmid, pCC110, which carries a wild-type version of *mreB*<sup>Gs</sup> in translational fusion with a 5' extension encoding a 6-histidine tag, was used as a template to generate pCC116, pCC117 and pCC115, carrying the *mreB*<sup>Gs</sup> gene deleted for codons 2–7 (FGIGTK), 102–105 (GLFA), or both, respectively. For this, pCC110 was PCR-amplified using primers cc582/cc583 (to generate pCC116) or cc584/cc585 (to generate pCC117) (**Supplementary file 5**) and the PCR products were treated with *Dpn*I prior to transformation into *E. coli*. To generate pCC115, isothermal assembly was performed with two PCR products generated using primers cc582/cc585 and cc583/cc584 and pCC110 as template, and the product was transformed into *E. coli*. Following extraction and sequencing, the four resulting pCC plasmids were transformed into the T7 *express E. coli* expression host (Table. S4).

The his-tagged proteins were produced in T7 express *E. coli* cells grown in LB broth supplemented with kanamycin. Expression of recombinant MreB was induced by the addition of IPTG at the final concentration of 1 mM, when cultures reached an optical density at 600 nm of 0.6. Expression was performed over-night at 15 °C, with maximum aeration. Bacteria were harvested by centrifugation (5000 *g* for 7 min at 4 °C).

Pellets were resuspended in buffer A (20 mM Tris pH 7, 500 mM KCl) supplemented with EDTA-free complete protease inhibitor (Roche) and 250 µg/mL of lysozyme. Cells were disrupted by sonication on a Vibra-Cell VC505 processor (Sonics & Materials, Inc, Newton, CT, USA) for 10 min with 10 seconds on/off cycles at 50% power, and the supernatant was collected after clarification by centrifugation (40 000 *g* for 20 min at 4 °C). The 6-histidine-tagged MreB variants followed a two-step purification procedure. The proteins were first purified by affinity chromatography on a Ni-nitrilotriacetic acid (Ni-NTA)

agarose resin (Thermo Fisher Scientific). The column was washed with buffer A supplemented with 20 mM imidazole, and proteins were eluted with a step gradient of imidazole (100 mM to 400 mM) in buffer A. The collected fractions were analyzed by electrophoresis, using a 12% polyacrylamide precast gel (Mini-PROTEAN TGX stain free, Bio-Rad). Fractions containing the purest form of the proteins were loaded on a size exclusion chromatography HiLoad 16/60 Superdex 200 pg column (GE Healthcare Life Sciences / Cytiva), pre-equilibrated with buffer B (buffer A supplemented with 1 mM DTT and 2 mM EDTA) connected to an AKTA FPLC system (GE Healthcare Life Sciences). Fractions corresponding to the elution peaks were analyzed by electrophoresis to assess the presence of MreB, pooled and concentrated with an ultrafiltration spin column (Vivaspin, 10 000 MWCO; Sartorius), up to a maximum of 0.5 mg/mL (13.4 µM), as determined from the absorption at 280 nm measured using a Nanodrop spectrophotometer (Thermo Fisher Scientific). The recombinant proteins were aliquoted and immediately frozen and stored at –20 °C.

## Preparation of lipid monolayers and negative stain electron microscopy

Unless stated otherwise, the standard reaction buffer supporting polymerization contained 20 mM Tris pH7, 100 mM KCl, 2 mM ATP, and 5 mM $Mg^2$, in which MreB was set to polymerize for 1–2 h at 25–37°C (**Supplementary file 2**). Polymerization of MreB on lipids was induced by creating a lipid monolayer on droplets containing MreB (typically 1.34 µM (0.05 mg/mL)) in the reaction buffer. Lipids from *E. coli* TE were dissolved to 2 mg/mL in chloroform in a glass vial and stored at –20 °C. Lipid preparations were diluted in chloroform to a final concentration of 0.5 mg/mL on the day of the experiment. Approximately 200 nL of lipid preparation were dropped on top of the droplets containing MreB in the reaction buffer, and incubated for polymerization.

For TEM observations, carbon-coated electron microscopy grids (CF300-Cu from Electron Microscopy Sciences or 1753-F from Ted Pella, Inc) were used without treatment to preserve their hydrophobic properties and allow an optimum adsorption of the lipid monolayer. To observe hydrophilic particles, grids were glow-discharged by a 30 s plasma treatment (PDC001, Harrick Plasma). Following polymerization, grids are placed, carbon side down, on top of lipid-coated reaction droplets and gently lifted after 2 min incubation. Grids were stained with either a solution of 2% uranyl formate or 1% uranyl acetate and air-dried prior to TEM observation. TEM images were acquired on a charge-coupled device camera (AMT) on a Hitachi HT 7700 electron microscope operated at 80kV (Milexia – France) or a Tecnai G2 LaB6 (Thermo Fisher FEI) microscope operated at 200 kV or a Tecnai Spirit (Thermo Fisher FEI) microscope operated at 80 kV.

Fourier Transformation of MreB sheets was done using ImageJ to obtain diffraction patterns. For 2D processing, a set of images was collected at a magnification of 50000 with a pixel size of 2.13 Å per pixel and a defocus varying from –2 to –1 µm, using a Tecnai G2 LaB6 (Thermo Fisher FEI) microscope operated at 200 kV and a F-416 TVIPS 4K4K camera. To obtain 2D class averages, particles were classified and aligned, using SPIDER (**Frank et al., 1996**). A total of 1554 particles were windowed out into 99x99 pixels images by using the Boxer interface of EMAN (**Ludtke et al., 1999**) and appended into a single SPIDER file, then normalized against the background. One round of reference-free alignment and classification was performed before references were selected from the first-class averages. Several rounds of multireference alignment and classification were then performed, and new references were selected from the class averages until no further improvement was obtained.

## Quantification of MreB filaments on EM images

We set up a workflow to quantitatively compare the ability of MreB to form polymers in different conditions from EM images. To circumvent the issue of the highly heterogeneous distribution of polymers on the EM grids, which could bias an analysis based on visual inspection, we acquired, for each experimental replica, images on 12 random locations covering the entire grid (**Figure 2—figure supplement 1**). For each field, the density of pairs of protofilaments observed was scored (none, '-'; low density, '+' or high density, '++').

## Preparation of liposomes and cryo-electron microscopy

*E. coli* TE was dissolved in chloroform, aliquoted, dried under a stream of argon, and desiccated for 1 h under vacuum. The liposome solution was made by resuspending desiccated TE in polymerization buffer (20 mM Tris-HCl pH 7, 100 mM or 500 mM KCl, 2 mM ATP, 5 mM $MgCl_2$) on the day of the

experiment, to a final lipid concentration of 0.1 mg/mL. 1.34 µM (0.05 mg/mL) of purified MreB was mixed with the liposome solution and incubated 2 hr at room temperature. Four µL of sample were applied to a glow-discharged holey lacey carbon-coated cryo-electron microscopy grids (Ted Pella, USA). Most of the solution was blotted away from the grid to leave a thin (<100 nm) film of aqueous solution. The blotting was carried out on the opposite side from the liquid drop and plunge-frozen in liquid ethane at −181 °C using an automated freeze plunging apparatus (EMGP, Leica, Germany). The samples were kept in liquid nitrogen and imaged using a Tecnai G2 (FEI, Eindhoven, Netherlands) microscope operated at 200 kV and equipped with a 4k4k CMOS camera (F416, TVIPS), or a Glacios 200kV (Thermo Fisher) microscope equipped with a Falcon IV direct detector. The imaging was performed at a magnification of 50 000 with a pixel size of 2.13 Å using a total dose of 10 electrons per Å$^2$.

## Preparation of liposomes and QCM-D measurements

DOPC and DOPG lipid mixtures were prepared in chloroform as described above except that desiccation was performed under vacuum overnight. The lipids were rehydrated in 10 mM Tris pH 7.0, 100 mM NaCl, 5 mM MgCl$_2$ buffer at a final concentration of 5 mg/mL using three consecutive cycles of freezing in liquid nitrogen and thawing in an ultrasonic bath (Merck). The rehydrated lipid solutions were extruded 21 times through a 100 nm diameter pore size polycarbonate membrane (Avanti Polar Lipids Inc). The extruded solutions were stored at 4 °C and consumed within a week after preparation.

A QCM-D E4 (QSense AB, Biolin Scientific AB, Gothenburg, Sweden) was used to measure MreB binding to planar supported lipid bilayers (SLBs) as previously reported for MinD and MinE (*Renner and Weibel, 2012*). Briefly, during QCM-D measurements, frequency and dissipation changes are recorded based on the piezoelectric properties of the crystal probe (*Rodahl et al., 1995*). The quartz crystals (QSense AB, Biolin Scientific AB, Gothenburg, Sweden) were coated with a custom 50 nm-thick layer of silicon dioxide by chemical vapor deposition (GeSiM GmbH, Dresden, Germany). Prior to each measurement, quartz crystals were thoroughly cleaned in a 1:1:5 volumetric ratio of concentrated ammonium hydroxide (Sigma-Aldrich), 30% hydrogen peroxide (Sigma Aldrich), and ultrapure water (Merck) at 70 °C for 3 min. Prior to liposome deposition, the quartz crystals were then placed and oxidized in a plasma cleaner (Harrick Plasma, Ithaca, NY) for 2 min at high radio frequency. The oxidized (activated) crystals were placed into the QCM-D measurement chambers and immediately covered with 10 mM Tris buffer, 100 mM NaCl and 5 mM MgCl$_2$. Subsequently, after a stable baseline was established, a liposome working solution (0.2 mg/mL) was pumped into the measurement chambers at 200 µL/min. After 2–20 min of incubation, a characteristic profile of supported planar lipid bilayer formation was observed (*Figure 3—figure supplement 3A*; *Keller et al., 2000*). After 5 min, the SLBs were rinsed with 10 mM Tris buffer containing 100 mM NaCl and 5 mM MgCl$_2$ to remove unbound vesicles at 100 µL/min. The buffer was next exchanged to the reaction buffer (20 mM Tris-HCl pH 7, 100 mM or 500 mM KCl, 1 mM DTT and 2 mM EDTA). After a stable baseline was observed, MreB (ATP, ADP, AMP-PNP) in reaction buffer (20 mM Tris-HCl pH 7, 100 mM or 500 mM KCl, 1 mM DTT, 5 mM MgCl$_2$) was added at 0.1, 0.5, 1 and 5 µM (low to high concentration) to the SLB at a pump speed of 100 µL/min for 5 min. The adsorption of MreB wild-type and mutants was measured for at least 20 min, until a plateau was approximately reached, before exchanging and rinsing with reaction buffer for 5 min at 100 µL/min. In a series of experiments (from low to high MreB concentration), MreB was almost completely displaced by the rinsing step, allowing multiple adsorption steps on a single SLB. However, at higher MreB concentrations the rinsing was only partially effective (*Figure 3—figure supplement 3B*). To avoid history effects on a SLB, we also reversed the MreB concentration steps (from high to low concentration). We calculated the thickness from frequency shifts using the Sauerbrey model included in the commercial analysis software tool QTools (QSense AB, Biolin Scientific AB, Gothenburg, Sweden). Each measurement was repeated at least twice with 2–3 repeats.

## Circular dichroism

The secondary structure of recombinant WT and mutant forms of MreB were analyzed by circular dichroism (CD). Far-UV spectra were recorded on a J-810 spectropolarimeter (Jasco). Spectra were recorded from 260 to 200 nm at 20 °C in 1 mm path-length quartz cuvette at a protein concentration of 10 µM in 50 mM NaPO$_4$ buffer at pH 7. Each CD spectrum was obtained by averaging 4 scans

collected at a scan rate of 200 nm/min. Baseline spectra obtained with buffer were subtracted for all spectra.

## NTPase activity assay

ATPase and GTPase activity of MreB were assayed by measuring the release of free inorganic phosphate ($P_i$) in a colorimetric assay using malachite green (*Kodama et al., 1986*; *Mao et al., 2017*). $P_i$ produced was measured after a fix (end-point) or various (kinetic) incubation times in the reaction buffer (20 mM Tris, 100 mM or 500 mM KCl, 5 mM $MgCl_2$) with appropriate supplements (e.g. 0.5 mM ATP or GTP, 0.05 mg/mL liposomes). The liposome solution was made on the day of the experiment by resuspending desiccated TE in water to 1 mg/mL. The reaction was initiated by the addition of MreB to the reaction mixture and ended by addition of 1 reaction volume of malachite revelation buffer (0.2% (w/v) ammonium molybdate, 0.7 M HCl, 0.03% (w/v) malachite green, 0.05% (v/v) Triton X-100). Incubations were performed at 53°C and 37°C for 1 h (end point) or less (kinetics). The quantity of $P_i$ produced was determined by measuring the absorbance at 650 nm on a 96-well plate spectrophotometer (Synergy 2, Biotek). A mock reaction devoid of protein constituted the blank. A standard curve was made with a range of $KH_2PO_4$ diluted in the reaction buffer.

## Crystallization, structure determination and refinement

Freshly purified MreB$^{Gs}$ containing an N-terminal 6xHis-tag (stored in 20 mM Tris pH7, 500 mM KCl, 2 mM EDTA, 1 mM DTT) was concentrated by centrifugation using a Vivaspin 5000 MWCO membrane tube (Sartorius). All crystallization assays were performed at 293 K by sitting-drop vapor diffusion using facilities from the crystallization platform of I2BC. Crystals of apo MreB$^{Gs}$ were obtained from a 100:100 nL mixture of protein at 3 mg/mL (80 µM) with a crystallization solution composed of 33% polyethylene glycol (PEG) 300 in 0.1 M MES pH 6.7. For co-crystallization assays, 10 mM ATP-Mg was added to 6 mg/mL (160 µM) of protein. Crystals of the complex were obtained with a crystallization solution containing 16% PEG 8000, 20% Glycerol and 0.04 M potassium phosphate. All crystals were flash-frozen in liquid nitrogen before data collection. Diffraction-quality crystals attained their full sizes in roughly 10–14 days.

Diffraction data were recorded on beam line PROXIMA 1 (synchrotron SOLEIL, France) at a wavelength of 0.9786 Å. Data were processed with the XDS package (*Kabsch, 2010*). All structures were solved by molecular replacement using the MOLREP program (*Vagin and Teplyakov, 1997*) using the crystal structure of MreB$^{Cc}$ (PDB ID 4CZJ; *van den Ent et al., 2014*), and the models were refined using PHENIX (*Liebschner et al., 2019*). The models were further improved by iterative cycles of manual rebuilding using COOT (*Emsley et al., 2010*). Final structural models were deposited in the Protein Data Bank (PDB; *Berman et al., 2000*). Statistics for all the data collections, refinement of the different structures and the PDB codes are summarized in *Supplementary file 1*. All structural figures were generated with PyMOL (The PyMOL Molecular Graphics System, version 1.2r3pre, Schrödinger, LLC n.d.). Protein structure comparison was performed using the PDBeFold service at European Bioinformatics Institute (http://www.ebi.ac.uk/msd-srv/ssm) (*Krissinel and Henrick, 2004*). Protein interfaces, surfaces and assemblies were analyzed using the PDBePISA service at European Bioinformatics Institute (http://www.ebi.ac.uk/pdbe/prot_int/pistart.html) (*Krissinel and Henrick, 2007*).

## Acknowledgements

We thank Davy Martin for CD acquisitions, Human Rezai and Juan Hermoso for useful discussions, and Xavier Henry for useful contributions upstream this work. This project has received funding from the European Research Council (ERC) under the European Union's Seventh Framework Program (FP7) and under the Horizon 2020 research and innovation program (grant agreement No 311231 and grant agreement No 772178, respectively, to RC-L). We also thank the Labex Cell(n)Scale (ANR-11-LABX0038), Paris Sciences et Lettres (ANR-10-IDEX-0001–02), and the Cell and Tissue Imaging (PICT-IBiSA), Institut Curie, member of the French National Research Infrastructure France-BioImaging (ANR-10-INBS-04). LDR acknowledges funding by the VolkswagenStiftung. This work benefited from the expertise of Christine Péchoux at the MIMA2 facility (Université Paris-Saclay, INRAE, AgroParisTech, GABI, 78350, Jouy-en-Josas, France) for TEM observations https://doi.org/10.15454/1.5572348210007727E12, and of the crystallization platform of I2BC, supported by the French Infrastructure for Integrated Structural Biology (FRISBI, ANR-10-INSB-05–05). We acknowledge the

synchrotrons ESRF (Grenoble, France) and SOLEIL (Saint-Aubin, France) for provision of synchrotron radiation facilities and we would like to thank the staffs of beamlines PROXIMA-2A and PROXIMA-1 at SOLEIL, and 1D23-2 at ESRF for assistance and advices during data collection.

## Additional information

### Funding

| Funder | Grant reference number | Author |
|---|---|---|
| European Research Council | ERC-SG | Rut Carballido-Lopez |
| European Research Council | ERC-CG | Rut Carballido-Lopez |
| Agence Nationale de la Recherche | ANR-11-LABX0038 | Aurélie Bertin |
| Agence Nationale de la Recherche | ANR-10-IDEX-0001-02 | Aurélie Bertin |
| VolkswagenStiftung | | Lars D Renner |
| Agence Nationale de la Recherche | ANR-10-INSB-05-05 | Sylvie Nessler |
| European Research Council | 311231 | Rut Carballido-Lopez |
| European Research Council | 772178 | Rut Carballido-Lopez |

The funders had no role in study design, data collection and interpretation, or the decision to submit the work for publication.

### Author contributions

Wei Mao, Formal analysis, Investigation, Visualization, Writing – original draft; Lars D Renner, Aurélie Bertin, Formal analysis, Validation, Investigation, Visualization, Methodology, Writing – review and editing; Charlène Cornilleau, Sana Afensiss, Sarah Benlamara, Yoan Ah-Seng, Investigation; Ines Li de la Sierra-Gallay, Formal analysis, Investigation, Visualization; Herman Van Tilbeurgh, Methodology; Sylvie Nessler, Data curation, Formal analysis, Supervision, Validation, Visualization, Writing – review and editing; Arnaud Chastanet, Conceptualization, Data curation, Formal analysis, Supervision, Validation, Investigation, Visualization, Methodology, Writing – original draft, Writing – review and editing; Rut Carballido-Lopez, Conceptualization, Formal analysis, Supervision, Funding acquisition, Validation, Writing – original draft, Project administration, Writing – review and editing

### Author ORCIDs

Wei Mao (ID) https://orcid.org/0000-0002-0443-651X
Lars D Renner (ID) http://orcid.org/0000-0003-0178-1788
Ines Li de la Sierra-Gallay (ID) https://orcid.org/0000-0003-2770-7439
Aurélie Bertin (ID) https://orcid.org/0000-0002-3400-6887
Arnaud Chastanet (ID) http://orcid.org/0000-0003-0320-4861
Rut Carballido-Lopez (ID) http://orcid.org/0000-0001-9383-8811

### Decision letter and Author response

Decision letter https://doi.org/10.7554/eLife.84505.sa1
Author response https://doi.org/10.7554/eLife.84505.sa2

## Additional files

### Supplementary files

• Supplementary file 1. Data-collection and refinement statistics.

- Supplementary file 2. List of polymerization conditions assayed.
- Supplementary file 3. List of proteins used in this study.
- Supplementary file 4. List of strains used in this study.
- Supplementary file 5. List of oligonucleotides used in this study.
- MDAR checklist

## Data availability

Protein structures data have been deposited in PDB under the accession codes 7ZPT and 8AZG.

The following datasets were generated:

| Author(s) | Year | Dataset title | Dataset URL | Database and Identifier |
|---|---|---|---|---|
| de la Sierra-Gallay IL, Mao W | 2023 | Crystal structure of MreB from Geobacillus stearothermophilus ATCC7953 | https://www.rcsb.org/structure/7ZPT | RCSB Protein Data Bank, 7ZPT |
| de la Sierra-Gallay IL, Mao W | 2023 | Crystal structure of MreB from Geobacillus stearothermophilus ATCC7953 in complex with ATP | https://www.rcsb.org/structure/8AZG | RCSB Protein Data Bank, 8AZG |

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
