## [Editor Report]

This important study makes the case that the assembly of MreB from Geobacillus, a Gram-positive organism differs substantially from MreB from the Gram-negative model organism, *Escherichia coli*. They make the compelling case that Geobacillus MreB assembly requires both interactions with membrane lipids and nucleotide binding: nucleotide hydrolysis is required for interaction with the membrane and interaction with lipids triggers polymerization. Altogether, these data make the strong case that MreB assembly dynamics can vary in significant, and organism-specific ways.

---

## [Decision Letter]

**Decision letter after peer review:**

Thank you for submitting your article "Polymerization cycle of actin homolog MreB from a Gram-positive bacterium" for consideration by *eLife*. Your article has been reviewed by 3 peer reviewers, including Petra Anne Levin as the Reviewing Editor and Reviewer #1, and the evaluation has been overseen by Suzanne Pfeffer as the Senior Editor. The following individual involved in the review of your submission has agreed to reveal their identity: Ethan Garner (Reviewer #2).

The reviewers have discussed their reviews with one another, and came to the conclusion that, while the current manuscript makes some important contributions--particularly with regard to structural data, it has limitations that will need to be addressed in a substantive and meaningful way in order to be reconsidered for publication in *eLife*.

*Reviewer #1 (Recommendations for the authors):*

1. Please write G+ and G- out as Gram positive and Gram negative (or gram positive and gram negative). Not only is this convention, but it also avoids confusion with G-actin, etc.

2. Line 334 "revelaed" is misspelled.

*Reviewer #3 (Recommendations for the authors):*

1. Interpretation of ligand density in the PDB 8ZAG (ATP-bound structure):

A figure of the electron density map showing the density of the ligand in the ATP structure should be shown.

The authors should re-visit the interpretation of the electron density at the nucleotide-binding pocket. Factors to recheck are as follows:

a. Is the third phosphate density really present? Given the high-resolution data and the redundancy, is there a possibility of observing the phosphate anomalous signal to confirm the presence of the third phosphate?

b. The third phosphate, even if present, has been possibly modeled in the wrong orientation. Please recheck.

c. Modelling of the Mg^2+^ ion is incorrect since none of the bond lengths is less than 2.4 A and the geometry of the coordination is incorrect. This is most probably a water molecule and not Mg^2+^. Interestingly this coincides with the position of a K^+^ ion in Spiroplasma MreB structure and a water molecule in other MreBs.

d. It is possible that the third phosphate has been built in place of the Mg^2+^? There are remnants of Fo-Fc density in the vicinity of the phosphates. A direct hydrogen bonding with aspartate and glutamate side chains is highly unlikely for the phosphate.

The authors are requested to check and correct the errors and resubmit the PDB file if required, before the release of the coordinates.

2. Could authors provide information on the ATP hydrolytic activity for the membrane binding motifs mutants, in the presence and absence of liposomes? It can also be that the mutants are able to undergo hydrolysis like wildtype without liposomes, but have lower (but not dead) ATPase activity in the presence of liposomes since you do see filaments in a few percentages of the fields for these mutants.

3. Line 306: Rather than just a possibility of nucleotide binding affecting the polymerization, did you go higher in MreB concentration? Did this make a difference? Since you are working in saturating concentrations of the nucleotide already an increase from 2 to 10 mM of the nucleotide is unlikely to make a difference.

4. Line 308: If filaments bound to ATP bind to the membrane, but AMPPNP or ADP do not bind, it is essential that the ADP-ALFx complex, the transition state analog, has to bind. The ATP in the presence of MreB will be transitioning through a mixture of states of AMPPNP, ADP-AlFx and ADP. Hence, a control with ADP-AlFx is essential to support this conclusion.

5. If the conclusion is that monomers cannot bind and assuming that AMPPNP and ADP states do not form polymers is true, then why is it that the monomeric proteins in AMPPNP and ADP states should bind as monomers to SLBs? How can the authors explain that this is not observed?

A concentration of 0.5 μm cannot be considered as monomers assuming 0.55 to be 100 % filaments. Hence, 0.5 μm and 1 μm considered for the binding study comparison is essentially the same data and do not compare monomer vs filament.

6. Line 349 – 351: Isn't this an indication that the readout on EM is membrane binding and not polymerization? How are the authors excluding the possibility that these do not form filaments, albeit at a higher critical concentration?

7. Lines 312 – 323: How does this data change for the monomeric binding? Monomer binding can also be demonstrated in the absence of ATP rather than testing at a lower polymeric concentration below 0.55.

8. Line 396: How do these observations suggest that polymerization and depolymerization are dynamic processes? Here since there is no change in density or length, it could also mean that they are static, and hydrolysis happens on the membrane-bound filaments.

Has the ATPase activity been tested in limiting concentrations of ATP? Then with time, there should be a disappearance of filaments from the lipids, if the sentence of steady state is indeed true.

Dynamics can be observed only if the experiment is repeated in limiting concentrations of ATP.

9. Line 940: How can error bars be included if these are only two independent repeats? Error bars require a minimum of three values for a statistically meaningful calculation.

10. A concentration of 1.37 μm MreB implies that the void fraction in SEC was used for the assay. Have the authors planned for an ATPase residue mutant to substantiate that the activity is indeed from the purified MreB?

11. In Line 221, (Figure 2—figure supplement 3B). "These deformed vesicles confirmed that MreBG ", replace MreBG to MreBGs.

12. In Figure 3 (Supplementary Figure 3), Under the mollicutes section, Spiroplasma helicoides, MreB (Change it to MreB1).

13. In Figure 1 figure supplement – 2, could the authors provide a calibration curve for the size exclusion profiles of the purified protein. Also, please include elution volume in the x-axis and not time.

14. The statement made in lines 441 – 442, that "Differences in the purity of the nucleotide stocks used in these studies could also explain some of the discrepancies" is not justified. For all the EM experiments, the amount of nucleotides used is in excess (so even if there is some level of impurity/degradation), the surplus amount (500 to 1000 fold excess) of nucleotide added should nullify the effect.

---

## [Author Response]

Reviewer #1 (Recommendations for the authors):1. Please write G+ and G- out as Gram positive and Gram negative (or gram positive and gram negative). Not only is this convention, but it also avoids confusion with G-actin, etc.

Done.

2. Line 334 "revelaed" is misspelled.

Thanks for pointing out this typo, this has been fixed.

Reviewer #3 (Recommendations for the authors):1. Interpretation of ligand density in the PDB 8ZAG (ATP-bound structure):A figure of the electron density map showing the density of the ligand in the ATP structure should be shown.The authors should re-visit the interpretation of the electron density at the nucleotide-binding pocket. Factors to recheck are as follows:a. Is the third phosphate density really present? Given the high-resolution data and the redundancy, is there a possibility of observing the phosphate anomalous signal to confirm the presence of the third phosphate?

We have checked and the electron density of the nucleotide is very clear and an omit map (Figure 4-S5) confirmed the presence of the third phosphate.

b. The third phosphate, even if present, has been possibly modeled in the wrong orientation. Please recheck.

We tried to refine the structure with the classical conformation of the nucleotide by inverting our γ-phosphate and our Mg^2+^ ion but it does not fit into the electron density map (see Author response image 1):

**Author response image 1. sa2fig1:** Electron density map after refinement with the classical conformation of ATP-Mg. The phosphate and the Mg ion are clearly in the negative Fo-Fc (in red) while the positive Fo-Fc map (in green) clearly coincide with our superimposed atypical conformation (in cyan).

c. Modelling of the Mg^2+^ ion is incorrect since none of the bond lengths is less than 2.4 A and the geometry of the coordination is incorrect. This is most probably a water molecule and not Mg^2+^. Interestingly this coincides with the position of a K^+^ ion in Spiroplasma MreB structure and a water molecule in other MreBs.

Thank you very much for this remark. We agree and we have replaced the Mg^2+^ ion by a water molecule.

d. It is possible that the third phosphate has been built in place of the Mg^2+^? There are remnants of Fo-Fc density in the vicinity of the phosphates. A direct hydrogen bonding with aspartate and glutamate side chains is highly unlikely for the phosphate.The authors are requested to check and correct the errors and resubmit the PDB file if required, before the release of the coordinates.

We now explain in the text that the atypical conformation of the observed ATP molecule is most probably due to the absence of magnesium in the crystal. In the absence of Mg^2+^, the β-phosphate can slightly rotate and bring the γ-phosphate in the atypical orientation observed in the electron density map. We have resubmitted the PDB file after correcting the errors.

2. Could authors provide information on the ATP hydrolytic activity for the membrane binding motifs mutants, in the presence and absence of liposomes? It can also be that the mutants are able to undergo hydrolysis like wildtype without liposomes, but have lower (but not dead) ATPase activity in the presence of liposomes since you do see filaments in a few percentages of the fields for these mutants.

We are thankful to reviewer #3 for this suggestion. We were making the same assumptions but we had not measured the ATPase activity of the mutants. We have now measured it in the presence and in the absence of liposomes as per the reviewer request, and added the results to the manuscript (Figure 5-S1D-E). As expected, the mutants retained basal ATPase activity (similar to the wild-type protein) in the absence of lipids, while in the presence of lipids they showed an increased activity that mirrors the amount of polymers quantified.

3. Line 306: Rather than just a possibility of nucleotide binding affecting the polymerization, did you go higher in MreB concentration? Did this make a difference? Since you are working in saturating concentrations of the nucleotide already an increase from 2 to 10 mM of the nucleotide is unlikely to make a difference.

We are afraid we do not fully understand this comment. L306 of our original manuscript corresponded to the QCM-D assay in the presence of ADP and AMP-PNP. All QCM-D assays were performed at higher MreB concentration already (5 µM), in addition to 0.5 and 1 µM. No binding was observed in the presence of ADP and AMP-PNP, for any of the MreB concentrations tested.

See responses to 1 and 2 above for the polymerization assays by TEM at higher MreB concentration.

4. Line 308: If filaments bound to ATP bind to the membrane, but AMPPNP or ADP do not bind, it is essential that the ADP-ALFx complex, the transition state analog, has to bind. The ATP in the presence of MreB will be transitioning through a mixture of states of AMPPNP, ADP-AlFx and ADP. Hence, a control with ADP-AlFx is essential to support this conclusion.

Thank you for this useful suggestion. On the light of our new data (filaments observed in the presence of ADP and AMP PNP), adding this experiment is no longer needed and beyond the scope of the current manuscript. We plan to address the question of the transition state in depth, a thorough analysis is needed for this.

5. If the conclusion is that monomers cannot bind and assuming that AMPPNP and ADP states do not form polymers is true, then why is it that the monomeric proteins in AMPPNP and ADP states should bind as monomers to SLBs? How can the authors explain that this is not observed?

This comment made us examine in detail our QCM-D data at 0.1 µM MreB (well below the critical concentration, and thus presumably unpolymerized MreB). Interestingly, a shift in frequency was detected even at 0.1 µM MreB in the presence of ATP but not of ADP or AMP-PNP, raising the intriguing possibility that MreB^Gs^-ATP monomers (or very short oligomers), but not MreB-ADP and MreB-AMP-PNP monomers, might interact with the membrane (Figure 3C and Figure 3-S3B, C, F). In the presence of ADP or AMP-PNP, we were not able to detect any significant binding at either low or high concentrations of MreB (Figure 3C and Figure 3-S3D, F). We also increased the concentration of ADP or AMP-PNP to exclude the possibility that the binding was simply affected by a decreased affinity of MreB^Gs^ for these nucleotides. Higher concentrations of ADP and AMP-PNP did not restore the binding of MreB^Gs^ to the SLBs neither (Figure 3-S3G).

Because binding was detected at 0.1 µM MreB in the presence of ATP but not of ADP and AMP-PNP, it is tempting to speculate that lipid binding might occur prior to polymerization into pairs of filaments on the membrane surface. However, we cannot exclude the presence of some MreB-ATP filaments in this QCMD experimental conditions even at the low 0.1 µM MreB concentration and thus we cannot draw any strong conclusion. We have reworked the text to remove any issuing interpretation of the QCMD data at 0.1 µM MreB but still point to the interesting observation that binding is observed at this low concentration of MreB only in the presence of ATP. We thank the reviewer for this useful comment.

A concentration of 0.5 μm cannot be considered as monomers assuming 0.55 to be 100 % filaments. Hence, 0.5 μm and 1 μm considered for the binding study comparison is essentially the same data and do not compare monomer vs filament.

At 0.55 µM (0.02 mg/ml) filaments are seen in ~10 % of the fields, versus 100% at 1.3 µM (0.045 mg/ml) (Figure 2B). However, as indicated in the previous answer, we realized that we cannot draw any strong conclusion on whether the residual binding at low MreB concentrations is due to monomers or filaments and we have therefore removed our interpretation of these data.

6. Line 349 – 351: Isn't this an indication that the readout on EM is membrane binding and not polymerization? How are the authors excluding the possibility that these do not form filaments, albeit at a higher critical concentration?

We agree with the reviewer that we could not discriminate between the two possibilities (lack of membrane binding vs lack of polymerization) on the EM experiments in the presence of lipids. We have now added complementary semi-quantitative EM experiments that show absence of polymers in solution too for the three mutants (Figure 4 -S4A).

7. Lines 312 – 323: How does this data change for the monomeric binding? Monomer binding can also be demonstrated in the absence of ATP rather than testing at a lower polymeric concentration below 0.55.

See response to 5 above. Binding of monomers in the QCM-D experiments cannot be conclusively demonstrated but our QCM-D results at 0.1 µM MreB-ATP (below the critical concentration in the presence of lipids) suggest that only MreB-ATP monomers (or a nucleated form of the filament) might bind to the membrane, while MreB-ADP or MreB-AMP-PNP monomers (or filaments) do not.

8. Line 396: How do these observations suggest that polymerization and depolymerization are dynamic processes? Here since there is no change in density or length, it could also mean that they are static, and hydrolysis happens on the membrane-bound filaments.Has the ATPase activity been tested in limiting concentrations of ATP? Then with time, there should be a disappearance of filaments from the lipids, if the sentence of steady state is indeed true.Dynamics can be observed only if the experiment is repeated in limiting concentrations of ATP.

We agreed with the referee that the ATPase findings were not conclusive. We have now performed the EM experiments in limiting conditions of ATP as suggested. In these conditions, the dual filaments disappeared from the lipid layer over time, providing further evidence for their dynamic nature (new Figure 5B). These new data have been added to the manuscript. Many thanks to reviewer #3 for this great suggestion to confirm the steady-state of the filaments!

9. Line 940: How can error bars be included if these are only two independent repeats? Error bars require a minimum of three values for a statistically meaningful calculation.

Error bars here represent standard deviations (SD) and show how scattered the values are. They are not meant to inform on whether the differences between the datasets are statistically significant or not (and that would be true even if n was >2), since only a statistical test can tell. When we wondered if two datasets were statistically different, we performed a statistical analysis (Figure 3-S1).

10. A concentration of 1.37 μm MreB implies that the void fraction in SEC was used for the assay.

We are afraid that we do not understand this statement. Only the fractions corresponding to the later peak (lower MW) in SEC was used in our experiments, far away from the void. Or does the reviewer suggest that at this concentration the protein would be above its critical concentration for aggregation? This is not the case, 1.37 µM correspond to 0.05 mg/ml, our standard condition for polymerization, and 10x below the critical concentration.

We have homogenized MreB concentrations to µM throughout the text and figures (previously both mg/ml and µM were used) to prevent any confusion.

Have the authors planned for an ATPase residue mutant to substantiate that the activity is indeed from the purified MreB?

We are currently lacking a good ATPase mutant, but this is planned for future studies.

11. In Line 221, (Figure 2—figure supplement 3B). "These deformed vesicles confirmed that MreBG ", replace MreBG to MreBGs.

Corrected. Thanks.

12. In Figure 3 (Supplementary Figure 3), Under the mollicutes section, Spiroplasma helicoides, MreB (Change it to MreB1).

Corrected.

13. In Figure 1 figure supplement – 2, could the authors provide a calibration curve for the size exclusion profiles of the purified protein. Also, please include elution volume in the x-axis and not time.

We have replaced the former X axis in time by elution volumes.

The calibration curve for the column has now been added as panel B of the same Figure 1-S2.

14. The statement made in lines 441 – 442, that "Differences in the purity of the nucleotide stocks used in these studies could also explain some of the discrepancies" is not justified. For all the EM experiments, the amount of nucleotides used is in excess (so even if there is some level of impurity/degradation), the surplus amount (500 to 1000 fold excess) of nucleotide added should nullify the effect.

We agree with the reviewer, we have removed this sentence.

References

Arino J, Ramos J, Sychrova H (2010) Alkali metal cation transport and homeostasis in yeasts. *Microbiology and molecular biology reviews* 74: 95-120

Bean GJ, Amann KJ (2008) Polymerization properties of the *Thermotoga maritima* actin MreB: roles of temperature, nucleotides, and ions. *Biochemistry* 47: 826-835

Cayley S, Lewis BA, Guttman HJ, Record MT, Jr. (1991) Characterization of the cytoplasm of *Escherichia coli* K-12 as a function of external osmolarity. Implications for protein-DNA interactions in vivo. *Journal of molecular biology* 222: 281-300

Dersch S, Reimold C, Stoll J, Breddermann H, Heimerl T, Defeu Soufo HJ, Graumann PL (2020) Polymerization of *Bacillus subtilis* MreB on a lipid membrane reveals lateral co-polymerization of MreB paralogs and strong effects of cations on filament formation. *BMC Mol Cell Biol* 21: 76

Eisenstadt E (1972) Potassium content during growth and sporulation in *Bacillus subtilis*. *Journal of bacteriology* 112: 264-267

Epstein W, Schultz SG (1965) Cation Transport in *Escherichia coli:* V. Regulation of cation content. *J Gen Physiol* 49: 221-234

Esue O, Wirtz D, Tseng Y (2006) GTPase activity, structure, and mechanical properties of filaments assembled from bacterial cytoskeleton protein MreB. *Journal of bacteriology* 188: 968-976

Gaballah A, Kloeckner A, Otten C, Sahl HG, Henrichfreise B (2011) Functional analysis of the cytoskeleton protein MreB from *Chlamydophila pneumoniae*. *PloS one* 6: e25129

Harne S, Duret S, Pande V, Bapat M, Beven L, Gayathri P (2020) MreB5 Is a Determinant of Rod-to-Helical Transition in the Cell-Wall-less Bacterium *Spiroplasma*. *Curr Biol* 30: 4753-4762 e4757

Kang H, Bradley MJ, McCullough BR, Pierre A, Grintsevich EE, Reisler E, De La Cruz EM (2012) Identification of cation-binding sites on actin that drive polymerization and modulate bending stiffness. *Proceedings of the National Academy of Sciences of the United States of America* 109: 16923-16927

Lacabanne D, Wiegand T, Wili N, Kozlova MI, Cadalbert R, Klose D, Mulkidjanian AY, Meier BH, Bockmann A (2020) ATP Analogues for Structural Investigations: Case Studies of a DnaB Helicase and an ABC Transporter. *Molecules* 25

Mannherz HG, Brehme H, Lamp U (1975) Depolymerisation of F-actin to G-actin and its repolymerisation in the presence of analogs of adenosine triphosphate. *Eur J Biochem* 60: 109-116

Mayer JA, Amann KJ (2009) Assembly properties of the *Bacillus subtilis* actin, MreB. *Cell motility and the cytoskeleton* 66: 109-118

Nurse P, Marians KJ (2013) Purification and characterization of *Escherichia coli* MreB protein. *The Journal of biological chemistry* 288: 3469-3475

Pande V, Mitra N, Bagde SR, Srinivasan R, Gayathri P (2022) Filament organization of the bacterial actin MreB is dependent on the nucleotide state. *The Journal of cell biology* 221

Peck ML, Herschlag D (2003) Adenosine 5 '-O-(3-thio)triphosphate (ATP-γ S) is a substrate for the nucleotide hydrolysis and RNA unwinding activities of eukaryotic translation initiation factor eIF4A. *Rna* 9: 1180-1187

Popp D, Narita A, Maeda K, Fujisawa T, Ghoshdastider U, Iwasa M, Maeda Y, Robinson RC (2010) Filament structure, organization, and dynamics in MreB sheets. *The Journal of biological chemistry* 285: 15858-15865

Rhoads DB, Waters FB, Epstein W (1976) Cation transport in *Escherichia coli*. VIII. Potassium transport mutants. *J Gen Physiol* 67: 325-341

Rodriguez-Navarro A (2000) Potassium transport in fungi and plants. *Biochimica et biophysica acta* 1469: 1-30

Salje J, van den Ent F, de Boer P, Lowe J (2011) Direct membrane binding by bacterial actin MreB. *Molecular cell* 43: 478-487

Schmidt-Nielsen B (1975) Comparative physiology of cellular ion and volume regulation. *J Exp Zool* 194: 207-219

Szatmari D, Sarkany P, Kocsis B, Nagy T, Miseta A, Barko S, Longauer B, Robinson RC, Nyitrai M (2020) Intracellular ion concentrations and cation-dependent remodelling of bacterial MreB assemblies. *Sci Rep-Uk* 10

van den Ent F, Izore T, Bharat TA, Johnson CM, Lowe J (2014) Bacterial actin MreB forms antiparallel double filaments. *eLife* 3: e02634

Whatmore AM, Chudek JA, Reed RH (1990) The Effects of Osmotic Upshock on the Intracellular Solute Pools of *Bacillus subtilis*. *Journal of general microbiology* 136: 2527-2535